# Identifying Physiological Stress Biomarkers for Prediction of Pork Quality Variation

**DOI:** 10.3390/ani10040614

**Published:** 2020-04-02

**Authors:** Nikola Čobanović, Sanja Dj Stanković, Mirjana Dimitrijević, Branko Suvajdžić, Nevena Grković, Dragan Vasilev, Nedjeljko Karabasil

**Affiliations:** 1Department of Food Hygiene and Technology, Faculty of Veterinary Medicine, University of Belgrade, Bulevar oslobodjenja 18, 11000 Belgrade, Serbia; mirjana@vet.bg.ac.rs (M.D.); brankos@vet.bg.ac.rs (B.S.); nevena.ilic@vet.bg.ac.rs (N.G.); vasilevd@vet.bg.ac.rs (D.V.); nedja@vet.bg.ac.rs (N.K.); 2Clinical Center of Serbia, Center for Medical Biochemistry, 11000 Belgrade, Serbia; sanjast2013@gmail.com

**Keywords:** acute-phase proteins, carcass quality, meat quality, minimal preslaughter stress, physiological stress biomarkers, standard marketing conditions

## Abstract

**Simple Summary:**

Prediction of technological and sensory pork quality—during a pig’s life or quickly after slaughter—is increasingly required by the pork industry in order to classify carcasses or primary cuts of carcasses for different production lines. Therefore, there is increasing demand for the development of accurate, reliable, time-efficient, non-invasive, real-time tools for predicting pork and carcass quality characteristics. Based on this, the aim of this study was to assess the potential use of various physiological stress biomarkers as indicators of carcass and meat quality traits in slaughter pigs subjected to the standard marketing conditions and to minimal stressful preslaughter handling. According to the results of this investigation, lactate dehydrogenase can be considered as a useful predictor of pork quality, while cortisol, alanine amino transferase, and albumin could be useful in prediction of carcass quality.

**Abstract:**

This study assessed the potential use of various physiological stress biomarkers as indicators of carcass and meat quality traits in 240 pigs subjected to the standard marketing conditions and minimal stressful antemortem handling using Pearson correlations. The most important pork quality traits (pH and temperature, water holding capacity, and color) had limited correlations with stress metabolites (lactate, glucose), stress hormones (cortisol, adrenocorticotropic hormone), stress enzymes (creatine kinase, aspartate amino transferase, alanine amino transferase), electrolytes (sodium, chloride), and acute-phase proteins (haptoglobin, C-reactive protein, albumin), indicating poor reliability in predicting pork quality. Albumin level was moderately positively correlated with live weight, hot carcass weight, cold carcass weight, and back fat thickness. Alanine amino transferase level was moderately positively correlated with live weight, hot carcass weight, and cold carcass weight. Cortisol level was moderately positively correlated with live weight, hot carcass weight, cold carcass weight, and back fat thickness, and moderately negatively correlated with the lean carcass content. Increased lactate dehydrogenase level was moderately correlated with decreased drip and cooking loss. In conclusion, lactate dehydrogenase could help pork producers predict pork quality variation, while cortisol, alanine amino transferase, and albumin could be useful in prediction of carcass quality.

## 1. Introduction

The day of slaughter has been identified as one of the most stressful stages in a pig’s life, which negatively affects pig health, welfare, carcass characteristics, and meat quality [1,2,3,4,5]. This is because pigs are inevitably exposed to handling, loading, transportation, unloading, adverse weather conditions, mixing with unfamiliar animals, and physical and emotional stress owing to the new accommodation [6]. Preslaughter stress influences body homeostasis and metabolism, resulting in alterations in the physiometabolic blood profile (electrolytes, hormones, metabolites, and enzymes), subsequently decreasing carcass and meat quality [7,8,9].

Stress metabolites, such as blood lactate and glucose, can be useful indicators for assessment of preslaughter stress, and their high blood levels indicate accelerated rate and greater extent of postmortem muscle metabolism, resulting in low muscle pH, while carcass temperature remains high, thus causing the higher prevalence of pale, soft, and exudative (PSE) pork [10,11]. It has been reported that adrenocorticotropic hormone (ACTH) [12] and cortisol [7] are useful indicators of the stress level experienced by pigs during the preslaughter period. Although several studies have investigated the relationship between cortisol concentration and pork quality, the results are not consistent. Several studies reported that the increase in cortisol level was associated with higher meat pH, darker meat color, lower drip loss, and higher occurrence of dark, firm, and dry (DFD) pork [13,14,15]. In contrast, other researchers reported that higher cortisol level was related to a faster muscle pH decline, lighter meat color, lower water holding capacity, bringing as a consequence the occurrence of PSE pork [7,16]. However, some authors stated that the cortisol concentration had no impact on the pork quality [17]. Stress enzymes, such as creatine kinase (CK), lactate dehydrogenase (LDH), aspartate amino transferase (AST) and alanine amino transferase (ALT), can be useful for monitoring the quality of preslaughter conditions, and, thus, may be of value in the identification of pork with undesirable quality traits [18,19,20,21]. Previous studies [18,19,20,21] reported that elevated stress enzymes levels were associated with high muscle pH, darker meat color, increased water holding capacity, and higher prevalence of DFD pork. Preslaughter stress can also cause an increase in electrolyte concentrations, such as potassium, sodium, chloride, and calcium. This could negatively influence pork quality, which is shown by decreased muscle pH and lower water holding capacity, increasing the tendency towards PSE meat [7,22]. It has been reported that acute-phase proteins (APPs) are a useful tool in the assessment of animal welfare during the preslaughter period [23,24,25,26]. The increase in APP levels, such as pig major acute-phase protein (Pig-MAP), haptoglobin, C-reactive protein (CRP), and serum amyloid A (SAA), have been observed after long and short transportation, as well as following isolation and changes in the food administration pattern [23,24,25,26]. However, only a few published papers have investigated the relationships between increased APPs and lower meat quality in terms of meat acidification and water holding capacity [27,28,29].

Prediction of pork quality in the slaughterline on the day of slaughter is of paramount importance from logistical and financial points of view. Non-invasive, quick, accurate, and reliable on-site prediction of pork quality would facilitate the distribution of whole carcasses or carcass cuts for fresh pork, different processing sectors (dry-cured ham or sausage production), or for meat branding (e.g., premium meat, meat products with protection of geographical indications, protected designations of origin, and guaranteed traditional specialty) according to their intrinsic characteristics and potential quality value. To date, several studies [10,11,12,13,14,15,16,17,18,19,20,21,22,23,24,25,26,27,28,29] have investigated physiological stress biomarkers as potential predictors of meat quality, as indicated previously. However, the high inconsistency of results reflects the complexity of meat quality traits, which are affected by multiple interacting factors, including genetic background, breeding, environmental factors, preslaughter conditions, as well as slaughtering procedures. Accordingly, to ensure high accuracy and reliability in the results, it may be important to investigate many different physiological stress biomarkers at the same time and determine the predictive capacity of the biomarkers on the combined analysis of all investigated physiological stress biomarkers together. Identification of such physiological stress biomarkers could help in prediction of pork quality at a time that decisions in the abattoir can be made (i.e., during the period between pig slaughtering and carcass chilling). This information could also help pig producers and meat companies to identify the animal welfare problem in preslaughter management and slaughtering practices, and to correct the cause, which in turn would improve pig welfare and carcass and pork quality. Therefore, the aim of this study was to assess the potential use of various physiological stress biomarkers as indicators of carcass and meat quality traits in slaughter pigs subjected to the minimal stressful antemortem handling and standard marketing conditions for Southeastern Europe.

## 2. Materials and Methods

### 2.1. Animals, Preslaughter Handling, and Slaughter Procedure

This study was conducted on 240 slaughter pigs (122 barrows and 118 gilts), at approximately six months old, with an average live weight of 110 ± 1.44 kg. All pigs were of the same genetics ((Yorkshire × Landrace) sows sired with Pietrain boars) and originated from the same large-scale commercial farm. The farm was a conventional farrow-to-finish herd practicing all-in–all-out management, with free-range (i.e., outdoor) sows and confined (i.e., indoor) weaners and fattening pigs. The farm has 1300 breeding sows and produces about 18,000 fattening pigs a year. Pigs were housed in a finishing barn on a fully slatted floor, in groups of 30 individuals per pen, with average space allocation of 1 m^2^ per pig. During the growing–fattening period, food and water were freely available 24 h a day. Pigs were kept in the established social groups over the entire growing–finishing stage. The pigs were fattened for 180 ± 5 days until they reached about 110 kg live weight. Pigs were not deprived of food or water before being sent for slaughter.

The pigs were monitored through deliveries in eight consignments of 30 pigs (two consignments per season). Pigs were slaughtered in months representing traditional seasonal environments in the Republic of Serbia: January and February (winter), April and May (spring), July and August (summer), and October and November (autumn). During each consignment, ambient temperature and relative humidity were monitored using a digital thermo-hygrometer (Testo 625, Testo AG, Lenzkirch, Germany) at unloading and during lairaging. The average ambient temperature and relative humidity at unloading were: −8.0 ± 2.1 °C and 50.35% ± 3.7% in winter, 16.5 ± 2.1 °C and 63.0% ± 14.1% in spring, 33.0 ± 2.8 °C and 80.4% ± 5.9% in summer, and 12.5 ± 2.1 °C and 66.4% ± 9.1% in autumn. The average ambient temperature and relative humidity during lairaging were 4.4 ± 1.7 °C and 53.0% ± 7.5% in winter, 18.4 ± 2.3 °C and 56.0% ± 9.9% in spring, 30.8 ± 1.8 °C and 81.6% ± 3.7% in summer, and 14.0 ± 2.8 °C and 61.4% ± 16.1% in autumn.

Pigs were loaded using a metal ramp (5 m length, slope ≤ 15°) in groups of 4–5 pigs on the same commercial single deck lorry by the same loading crew and driver between 05:00 and 06:00 on the morning of each shipping day. Pigs were mostly handled with PVC sorting boards, while electric prods were only used when it was absolutely necessary. Average loading time was of 53 ± 15.4 min. Lorries departed from the farm immediately after loading. Pigs were transported by the same driver using the same commercial single deck lorry with two compartments. A rotation of the group position in the lorry according to the slaughter season was done for every load to avoid the confounding effect of the lorry compartment on the pig responses to transportation. This compartment rotation allowed one group of pigs per slaughter season to be transported in each lorry compartment. The lorry had natural ventilation, whereby side panels were 100% open during all seasons. During transportation, the lorry was not bedded and showers were not available. Pigs had no access to food or water in the lorry. Transportation time lasted about two hours at a mean speed of 60 km/h, with an average stocking density of 0.38 m^2^/pig. Upon arrival at the abattoir, lorries waited 9.13 ± 3.5 min on average, whereas unloading took 15.63 ± 6.9 min on average. Pigs were unloaded using a metal ramp (5-m length, slope ≤ 15°), weighed, and kept in roofed lairage pens for 3 h in groups of 30 individuals per pen (stocking density of 0.67 m^2^/pig). Lairage pens (5.0 × 4.0 × 2.0 m, width × length × height) had concrete floors and walls and a solid metallic gate. Lairage pens were not equipped with sprinkling systems; the ambient conditions were regulated by natural ventilation. During lairaging, food was not provided but water was freely available via two drinking nipples (one nipple for 15 pigs, with a flow rate of 1 L/min). Total fasting time, including the transportation and lairage times, was approximately six hours.

During the preslaughter period, pigs were subjected to the standard marketing conditions (preslaughter handling, transportation, and lairaging). Considering that the Republic of Serbia is a relatively small country, as are many other countries in Southeastern Europe, abattoirs are located close to the pig farms, so standard market conditions include short transportation (between two and four hours) and lairaging times (between one to three hours). In addition, mixing of pigs from different farms is not allowed and pigs should be handled gently. Gentle handling consists of moving pigs at a slow and calm pace with a PVC sorting board and rattle paddle, and without use of sticks. Electric prods can only be used as a moving tool of last resort when absolutely necessary on the hindquarters of adult pigs, whereby the duration of the shock should not exceed one second. Handling of pigs during the day of slaughter was performed under minimal stressful antemortem conditions: (i) loading and unloading (in groups of 4–5 pigs) from the transport vehicle to the lairage pens, as well as driving pigs from the holding pens to the stunning area, were carried out gently with a PVC sorting board (gentle pressure to the pig’s hindquarters and flank) and without use of electric prods or sticks; (ii) mixing of unfamiliar pigs did not occur at any preslaughter phases, as the loaded pigs came from one pen, after which they were kept in a lairage following the same distribution as in the lorry. Pigs were not separated by gender during transportation and lairaging. Pigs were inspected by official veterinary inspectors during lairaging for clinical symptoms of disease. The pigs in the current study were clinically normal. Pig slaughter and carcass processing were performed in compliance with the standard industry-accepted practices at the same small-scale commercial abattoir, with a yearly throughput of 22,000.

### 2.2. Physiological Stress Biomarkers

Blood samples from each pig were collected in a plastic cup from a bleeding wound at exsanguination and immediately transferred to two types of tubes: the first tube was treated with potassium oxalate/sodium fluoride to inhibit further glycolysis, and the second tube was treated with EDTA for anti-coagulation. Both types of tubes were inverted gently eight times immediately after collection. Blood lactate and glucose concentrations were determined using handheld devices (blood glucose: GlucoSure AutoCode, ApexBio, Taiwan; blood lactate: Lactate Scout, EKF Diagnostics, Magdeburg, Germany) from the first tubes (tubes treated with potassium oxalate/sodium fluoride). During sampling and prior to analysis with handheld devices, tubes treated with potassium oxalate/sodium fluoride were kept on ice. All measurements were completed within 10 min of exsanguination at the abattoir. After blood collection, the EDTA tubes were kept in shaved ice in a cooler box until centrifugation at 3000 rpm for 10 min within 2–4 h of collection. Plasma was transferred into 3 × 1.5 mL tubes and frozen at −20 °C until further analysis. All samples were analyzed within one month of collection. The plasma aliquots were analyzed for cortisol and ACTH using automated analyzer (Roche Cobas e601, Roche Diagnostics, Mannheim, Germany). The plasma aliquots were analyzed for haptoglobin, CRP, albumin, potassium, sodium, chloride, calcium concentrations, and enzymatic activities of CK, LDH, AST, and ALT using an automated analyzer (Architect c8000, Abbott, Wiesbaden, Germany). All samples were measured in triplicate, and values were averaged for analysis. The intra-assay coefficients of variation were 4.96%, 6.12%, 3.96%, 4.18%, 5.36%, 4.31%, 4.88%, 3.91%, 6.21%, 5.11%, 6.56%, 4.44%, and 4.98% for cortisol, ACTH, haptoglobin, CRP, albumin, potassium, sodium, chloride, calcium, CK, LDH, AST, and ALT, respectively.

### 2.3. Carcass Quality Measurements

Carcasses were clearly labeled with a carcass ticket to ensure that they originated from the 240 slaughter pigs studied. Immediately after splitting and final washing, carcasses (with head, tail, and feet) were weighed on a balance scale (Pig Carcass Automatic Weighing Scale, Qingdao Empire Machinery Co., Ltd, Shandong, China) (accuracy of 0.1 kg) to determine the hot carcass weight and re-weighed 24 h after chilling to obtain the cold carcass weight. Back fat and loin muscle (*Musculus longissimus dorsi*) thicknesses were measured in millimeters with a stainless steel ruler (stainless steel ruler model 4153000, Wolfcraft, Kempenich, Germany) (accuracy of 1.0 mm) at 8 cm from the midline of split carcasses, between the third and fourth ribs. Back fat thickness was taken as the minimum fat thickness of the visible fat, including rind covering the *Musculus gluteus medius*. Loin muscle thickness was measured at the shortest connection between the front (cranial) end of the *Musculus gluteus medius* and the upper (dorsal) edge of the vertebral canal. The lean meat content (%) was calculated using Zwei–Punke Messverfahren (ZP) method [30] based on the thickness of the back fat and loin depth, according to the following formula: y = 65.93356 − 0.17759 × (x1) + 0.00579 × (x1) − 52.54737 × (x1/x2); where y = estimated lean meat content of the carcass (kg), x1 = back fat depth (mm), and x2 = loin muscle depth (mm). This formula is valid for hot carcasses weighing between 60 kg and 120 kg. The numbers of skin lesions on the left side of the carcasses were counted by two trained observers in the chilling room 45 min postmortem using the Welfare Quality^®^ (Lelystad, The Netherlands) protocol [31], as described in Čobanović et al. [32].

### 2.4. Pork Quality Measurements

The pH and temperature of the *Musculus longissimus dorsi* muscles were measured 45 min and 24 h after slaughter between 10th and 11th ribs using a pH meter (Testo 205, Testo AG, Lenzkirch, Germany) with a digital identification system, temperature compensation sensor, and a proper glass electrode. The pH meter was calibrated with pH 4.00 and 7.00 phosphate buffers before each series of measurements and the electrode was rinsed with distilled water between each measurement. Values of pH and temperature were both measured in triplicate, and the average of the three measurements was taken as a final result. At the same anatomical location where pH and temperature were measured, two boneless loin samples (each 2.54 cm thick, ~100 g) were taken from each selected carcass, weighed on a semi-analytical scale, and used for color, marbling, and water holding capacity measurements. Instrumental and sensory color, as well as marbling, were determined at 24 h postmortem, after a standard 30 min blooming period [33]. Instrumental color was measured using a portable colorimeter (Konica-Minolta, Chroma Meter CR 410, Osaka, Japan) equipped with a 25 mm aperture, 0° viewing angle, and D65 illuminant. Before each series of measurements, the instrument was calibrated using a white ceramic tile. Measurements were made at six different random reading points on the surface of the loin muscle and in the core after slicing in order to obtain a representative average value of the color. The average L*, a*, and b* values of six measurements were taken as a final result. An analytical panel of three trained members assessed sensory color and marbling of meat samples by using the scaling method based on the National Pork Producer Council [34] color and marbling standards. Color scores ranged from 1 (pale color) to 6 (dark color), and marbling scores ranged from 1 (1% intramuscular fat content) to 10 (10% intramuscular fat content). Water holding capacity was measured using three methods: drip loss, thawing loss, and cooking loss. Drip loss measurements were performed based on the bag method [33], with the samples weighed individually and placed in a net enclosed in a polyethylene bag under atmospheric pressure, ensuring that the sample was not in contact with the bag. At the end of the 48 h storage period at 4 °C, the samples were removed from the bag, surface moisture was carefully dabbed with tissue paper, and then re-weighed. Drip loss was expressed as a percentage of the initial sample weight. After drip loss measurement, the same meat samples were placed in a plastic freezer bag and frozen at −20 °C. Afterwards, the samples were thawed at room temperature for 12–16 h and then taken from the bag, gently blotted dry with tissue paper, and re-weighed. Differences of weights gave the thawing loss of the samples [28]. The thawed samples were put in a plastic (Ziploc) bag and then placed in a continuously boiling water bath until the internal temperature reached 75 °C, which was measured using a thermometer with a hand probe (Testo 110, Testo AG, Lenzkirch, Germany) [33]. When the end point temperature was attained, the bags were removed from the water bath. Thereafter, the samples were cooled in an ice slurry and kept under chilled conditions (1–5 °C) until equilibration. The cooled samples were taken from the bag, gently blotted dry with tissue paper, and then re-weighed. Cooking loss was estimated by weighing them before and after cooking. Rigor mortis intensity was determined on the left carcass side 45 min postmortem by measuring the degree of angle between the body axis and foreleg according to Dokmanović et al.’s method [13]. For this purpose, photographic images of carcasses were taken at a distance of approximately 2 m and a height of 160 cm, parallel to the plane in which the carcasses were held. The angle was calculated in AutoCAD program. Angle size and rigor intensity are inversely proportional (i.e., a smaller angle means a higher degree of rigor mortis). Subjective assessments of the rigor development were made 45 min postmortem on the *Musculus semimembranosus* in the split carcass using a three-point scale according to Čobanović et al. [32]: (1) muscle not in rigor; (2) muscle partly in rigor; and (3) muscle in full rigor. Semimembranosus muscle is normally exposed on the carcass medial surface and can be assessed by gauging its surface firmness using finger pressure, whereby in a muscle not in rigor the surface feels soft, while a muscle in rigor feels quite firm. The intra-measurement coefficients of variation for meat quality parameters were below 10%. Pork quality classes (pale, soft, and exudative—PSE; red, soft, and exudative—RSE; red, firm, and nonexudative—RFN; pale, firm, and nonexudative—PFN; dark, firm, and dry—DFD) were determined using pH values measured 24 h postmortem, drip loss variations, and light reflectance (L*), according to Koćwin-Podsiadła et al. [35] (Table 1).

### 2.5. Statistical Analysis

Statistical analysis of the results was conducted with SPSS software (Version 23.0, IBM Corporation, Armonk, NY, USA) [36]. The data were initially screened for linearity, normality of residuals (Shapiro–Wilk and Kolmogorov–Smirnov test), outliers, and homogeneity of variance (Levene’s test), and successfully passed all tests. Pearson’s correlation test was used to establish if there were any correlations between physiological stress biomarkers and carcass and meat quality parameters. Correlations were considered weak at |r| < 0.35, moderate at 0.36 ≥ |r| < 0.67, and strong at |r| ≥ 0.68 [37]. One way analysis of variance (ANOVA) was performed to detect significant differences of various physiological stress biomarkers between different pork quality classes. Significant means at *p* ≤ 0.05 were further compared using Tukey’s test (multiple comparisons). Slaughter season was added as a random factor to the model for the data analysis in order to control for the possible effects of ambient conditions on levels of physiological stress biomarkers. ANOVA was performed to test the effects of slaughter season on the physiological stress biomarkers. Pork quality classes and slaughter season served as independent variables, while physiological stress biomarkers served as dependent variables. Data were described by descriptive statistical parameters as the mean value, standard deviation, and minimum and maximum range. Potassium and calcium were not considered for statistical tests because their concentrations were above and below limits of detection, respectively. Each individual pig was considered an experimental unit. A probability level of *p* ≤ 0.05 was chosen as the limit for statistical significance in all tests.

## 3. Results

### 3.1. Relationships between Stress Metabolites and Carcass and Meat Quality of Slaughter Pigs

Relationships between stress metabolites and carcass and meat quality of slaughter pigs are shown in Table 2, Table 3, Table 4. Weak negative correlations were found between lactate and glucose levels and meat pH measured 45 min and 24 h postmortem; redness (a* value); and sensory color score (*p* < 0.05). In addition, lactate and glucose levels were weakly positively correlated with lean meat content, meat temperature measured 45 min postmortem, drip loss, cooking loss, and lightness (L* value) (*p* < 0.05). PSE pork had the highest (*p* < 0.05) lactate and glucose levels.

### 3.2. Relationships between Stress Hormones and Carcass and Meat Quality of Slaughter Pigs

Relationships between stress hormones and carcass and meat quality of slaughter pigs are shown in Table 2, Table 3, Table 4. Moderate positive correlations were found between cortisol level and live weight, hot carcass weight, cold carcass weight, back fat thickness, and marbling (*p* < 0.05). A moderate negative correlation was found between cortisol level and lean meat content (*p* < 0.05). In addition, cortisol level was weakly positively correlated with skin lesion score (*p* < 0.05). There were no significant differences (*p* > 0.05) between pork quality classes for cortisol and ACTH levels.

### 3.3. Relationships between Electrolytes and Carcass and Meat Quality of Slaughter Pigs

Relationships between electrolytes and carcass and meat quality of slaughter pigs are shown in Table 2, Table 3, Table 4. Electrolytes had limited correlations with carcass and meat quality parameters (*p* > 0.05). There were no significant differences between pork quality classes for sodium and chloride levels (*p* > 0.05).

### 3.4. Relationships between Stress Enzymes and Carcass and Meat Quality of Slaughter Pigs

Relationships between stress enzymes and carcass and meat quality of slaughter pigs are shown in Table 5, Table 6, Table 7. Moderate positive correlations were found between ALT level and live weight, hot carcass weight, and cold carcass weight (*p* < 0.05). In addition, CK, LDH, and ALT levels were weakly positively correlated with the loin muscle thickness and lean meat content (*p* < 0.05). LDH level was moderately negatively correlated with drip loss and cooking loss (*p* < 0.05). DFD pork had the highest concentrations of CK, LDH, and AST (*p* < 0.05).

### 3.5. Relationships between Acute-Phase Proteins and Carcass and Meat Quality of Slaughter Pigs

Relationships between acute-phase proteins and carcass and meat quality of slaughter pigs are shown in Table 5, Table 6, Table 7. Moderate positive correlations were found between albumin level and live weight, hot carcass weight, cold carcass weight, and back fat thickness (*p* < 0.05). In addition, albumin level was weakly positively correlated with loin muscle thickness and lean meat content (*p* < 0.05). A weak positive correlation was found between haptoglobin level and meat pH value measured 45 min and 24 h postmortem (*p* < 0.05), while weak negative correlations were found between haptoglobin level and drip loss, cooking loss, and lightness (L* value) (*p* < 0.05). DFD pork had the highest haptoglobin level (*p* < 0.05).

### 3.6. Interrelationships (r) between Physiological Stress Biomarkers in Slaughter Pigs

Interrelationships (r) between physiological stress biomarkers in slaughter pigs are depicted in Table 8. Electrolytes such as sodium and chloride levels were strongly positively correlated with each other (*p* < 0.05), and both were moderately positively correlated with albumin levels (*p* < 0.05). Albumin level was moderately positively correlated with cortisol and AST levels (*p* < 0.05). A weak positive correlation was found between albumin level and CK level (*p* < 0.05). Haptoglobin level was weakly positively correlated with cortisol, CK, LDH, and AST levels (*p* < 0.05). CK level was moderately positively correlated with LDH and ALT levels (*p* < 0.05), while LDH and AST levels were moderately positively correlated with each other (*p* < 0.05).

### 3.7. Relationships between Slaughter Season and Physiological Stress Biomarkers in Slaughter Pigs

Relationships between slaughter season and physiological stress biomarkers in slaughter pigs are shown in Table 9. Pigs slaughtered in summer had the highest lactate, sodium, chloride, and albumin levels (*p* < 0.05), but the lowest cortisol levels (*p* < 0.05). On the other hand, pigs slaughtered in winter had the highest ACTH levels (*p* < 0.05). The highest levels of CK and LDH were detected in pigs slaughtered in winter and summer (*p* < 0.05).

## 4. Discussion

The present investigation revealed that the lactate level was weakly positively correlated with the loin muscle thickness and lean meat content, but weakly negatively correlated with back fat thickness. In addition, glucose concentration was weakly positively correlated with lean meat content. Although the prevalence of the stress-susceptible n allele in pigs was not evaluated in this study, more than a third of the examined slaughter pigs (36.25%) were heterozygous (Nn genotype) in the ryanodine receptor (RYR1) gene (unpublished data). Genetic selection of pigs for improved meatiness has led to increased numbers of white muscle fibers that are extremely rich in glycogen, which resulted in two to three times higher muscle glycogen content in stress-carrier pigs (Nn genotype) than in stress-resistant pigs (NN genotype) [32,42]. This can explain a greater potential for accumulating lactate and glucose in the circulation of pigs with a higher percentage of meat [10,11]. Activation of the first stage of the stress response (short term, acute stress) stimulates the sympathetic–adrenal–medullary (SAM) axis, causing the release of catecholamines, including noradrenaline and adrenaline. Adrenergic stress response increases blood lactate and glucose levels through rapid muscle and hepatic glycogenolysis [20,29]. Consequently, skeletal muscles of acutely stressed pigs show a sharp drop in pH in the first 45 min postmortem, which in combination with high meat temperature induces denaturation of sarcoplasmic and myofibrillar proteins and reduction in their water holding capacity, subsequently resulting in pork with PSE characteristics [7,8]. This is partially confirmed by the present study, where elevated lactate and glucose levels weakly correlated with increased meat acidification (decreased pH_45min_ and pH_24h_, but increased T_45min_), reduced water holding capacity (increased drip and cooking loss), paler pork (increased L* and b* values, but decreased sensory color scores) and more developed rigor mortis (smaller foreleg angle and higher rigor score). The small and inconsistent correlations observed between stress metabolites and meat quality parameters indicates that they have limited use as predictors of the rate and extent of postmortem metabolism and final pork quality at slaughter. The level of preslaughter stress required to negatively affect pork quality traits should probably be greater than that needed to affect lactate and glucose levels in the bloodstream [43]. Another possible explanation is that the use of whole blood for lactate and glucose analysis results in an underestimation of their concentrations in circulation [43].

In this study, cortisol level was moderately positively correlated with live weight, hot carcass weight, cold carcass weight, and back fat thickness, but moderately negatively correlated with the lean carcass content, indicating that higher cortisol concentrations increase body fatness and decrease meatiness on a long-term basis [14]. These results reflect general metabolic effects of cortisol that affects the metabolism of fats and proteins by favoring the storage of fat (in the presence of insulin) at the expense of decreased synthesis and increased protein degradation. This occurs via peripheral catabolism and hepatic neoglucogenesis [10,13,14]. In addition, higher cortisol concentrations correlated with higher pork marbling, supporting the theory that cortisol contributes to storage of adipose tissue, not only under the skin and around organs, but also between and inside the muscle fibers [13]. It is likely that the same mechanism controls fat deposition regardless of location in the body, since previous studies [13,44] have reported a positive relationship between back fat thickness and pork marbling. The obtained results revealed that skin lesion score tended to increase with increasing cortisol level. Although the skin lesion type on pig’s carcasses was not evaluated in this study, 31.25% of the examined slaughter pigs had carcass lesions caused by fighting (unpublished data). In spite of standard marketing conditions and minimal stressful antemortem handling, half of the pigs (slaughtered in summer and winter) were exposed to adverse weather conditions on the day of slaughter, which have been demonstrated to be very strong stressors that could provoke fighting behavior in pigs [45,46]. Fighting behavior increases cortisol concentrations and results in carcass lesions [47]. However, the low magnitude of the correlation between cortisol level and skin lesion score found in this investigation can be ascribed to the cleaning and processing of the carcasses on the slaughterline, which presumably caused confluence of individual lesions and altered the count of skin lesions on pig carcasses that could be seen in live pigs [47]. The second stage of stress (long-term, chronic stress) activates the hypothalamic–pituitary–adrenal (HPA) axis, whose activation stimulates the secretion of corticotrophin-releasing factor from the hypothalamus, ACTH from the pituitary gland, and cortisol from the adrenal cortex [20]. The secretion of cortisol into the circulation induces the catabolic activity in peripheral tissues (glycogenolysis, proteolysis, and lipolysis) and anabolic activity in the liver (gluconeogenesis and protein synthesis) in order to increase the blood glucose concentration and provide the energy necessary to cope with the stressor [48]. This can rapidly accelerate muscle glycogenolysis and muscle glycogen depletion due to long-term stress during the preslaughter period, leading to lower production of lactic acid postmortem and resulting in pork with DFD characteristics [20]. Partially supporting this hypothesis, higher cortisol levels weakly correlated with increased water holding capacity of pork (decreased drip and cooking loss), while elevated ACTH levels weakly correlated with increased initial meat pH, darker pork color (decreased L* and b* values), and increased water holding capacity (decreased drip and cooking loss). Despite the fact that an increase in cortisol and ACTH levels increases the likelihood of DFD meat, there were no significant differences between pork quality classes for cortisol and ACTH levels. Regardless of the pork quality class, cortisol levels were much higher than the basal levels of the species. Accordingly, the results obtained in this investigation confirmed that cortisol and ACTH have limited uses as indicators of pork quality, which can be explained by the fact that several factors of variation (e.g., circadian rhythm, breed, sex, age, feeding, susceptibility to stress, and repeatability for the same stressor) may influence their concentrations [7]. Although the HPA axis responds more slowly to stressors than the SAM axis, cortisol concentration typically increases within 30 min after stressful situations and returns to the basal level after about three hours, indicating that this physiological stress biomarker is not very informative for the detection of chronic or “long-term” stress (lasting days, weeks, months, or even years, instead of hours) [49].

Albumin and sodium levels, both of which are indicators of hydration, were weakly positively correlated with the CK and ALT levels, indicating a higher degree of dehydration in exhausted pigs and in pigs with greater muscle damage [20]. Likewise, albumin and sodium levels were weakly positively correlated with cortisol level, implying that pigs subjected to long-term stress experienced higher degrees of dehydration as a consequence of food and water deprivation on the day of slaughter, especially during hot weather conditions. These results can be attributed to the fact that pigs slaughtered in summer were subjected to extremely high temperatures (29.5 °C to 35.0 °C) and high relative humidity (76.2–84.5%), which in combination with food and water deprivation during transportation and lairaging (six hours in total) caused higher degrees of dehydration. As supporting evidence of this notion, the highest levels of albumin, sodium, and chloride were detected in pigs slaughtered in summer. Even though some degree of dehydration is an unavoidable consequence during the preslaughter period due to food and water deprivation, this can be further exacerbated by vigorous physical exercise [20]. Thus, good hydration is of paramount importance for muscle glycogen reservoirs and presumably for muscle tissue damage [20]; therefore, water should be made more readily available in order to maximize pig welfare [29]. In this study, electrolytes showed limited correlations with carcass and meat quality traits, as well as with other physiological stress biomarkers, indicating they cannot be regarded as objective predictors of pork quality.

In this study, ALT level was moderately positively correlated with live weight, hot carcass weight, and cold carcass weight. In addition, loin muscle thickness and lean meat content tended to increase with increasing CK, LDH, AST, and ALT levels. The increase of stress enzymes activity with lean meat content can be attributed to the fact that skeletal muscles are the main sources of CK, LDH, AST, and ALT [11]. In this study, CK level was weakly positively correlated with a skin lesion score. Increases in circulating CK, LDH, AST, and ALT levels can be seen in cases of muscular damage, which are probably associated with a higher proportion of injuries due to loading, transportation, unloading, lairaging, adverse weather conditions, and vigorous physical exercise [18,29,50,51]. During strenuous muscle activity and muscular damage, CK, LDH, AST, and ALT are released into the bloodstream as a consequence of disruptions in the muscle cell membrane and cell permeability [18]. Increased intracellular energy demands during strenuous muscle activity, in combination with adrenergic activation of glycogenolysis, contributes to the muscle glycogen exhaustion, resulting in increased pH values and water holding capacity and darker color, as a consequence causing the occurrence of DFD meat [8,19,20]. This is partially confirmed by the present study, where elevated LDH levels moderately correlated with increased water holding capacity, as shown by the decreased drip and cooking loss. However, the increases in the concentrations of CK, LDH, and ALT weakly correlated with higher initial and final meat pH, decreased cooking loss, and darker pork color (decreased L* value). Furthermore, the highest (*p* < 0.05) mean CK, LDH, and AST levels were recorded in DFD pork. LDH concentrations in all pork quality classes were above the basal levels of the species, with the exception of the RFN pork, where LDH level was in the ranges considered normal. As mentioned before, more than a third of the examined slaughter pigs (31.25%) had fighting-type bruises on their carcasses, probably as a consequence of aggressive behavior caused by adverse weather conditions on the day of slaughter. During this investigation, pigs slaughtered in summer and winter were exposed to extreme weather conditions, such as high relative humidity (76.2–84.5%) and environmental temperatures above (29.5 to 35.0 °C) or below (−9.5 to −5.5 °C) the thermoneutral zone for slaughter weight pigs (15–25 °C, [4]). When the environmental temperature exceeds the upper threshold of thermal tolerance, the resting time is interrupted and pigs start to search for cool areas to lie down without contact with other individuals [46]. If this is not possible, pigs become agitated, which increases aggression between pen-mates [46]. When the environmental temperature falls below the lower threshold of thermal tolerance, pigs tend to maintain body temperature by lying in close proximity to each other to create a warmer microclimate [4,45]. However, grouping of pigs decreases the space allocation, which increases fights between pen-mates trying to find a place to rest [4,45]. Therefore, both heat and cold stress could provoke fighting behavior in pigs, resulting in carcass lesions, muscle damage, and pork quality deterioration. This finding is further supported by the fact that the highest CK and LDH levels were recorded in pigs slaughtered in winter and summer, which confirmed that levels of stress enzymes increase as a consequence of heat and cold stress [1,18]. Considering the obtained results, only LDH can be considered a useful predictor of pork quality at slaughter, while ALT could be used as a useful predictor of carcass quality. However, even though ALT is released in the bloodstream following muscle damage, this enzyme is not tissue specific and could also indicate hepatic disorders; therefore, it must be interpreted together with CK (only released from myocardial and skeletal muscle) and LDH (found in the skeletal muscle in the sarcomeres) to be a reliable predictor of carcass quality [18,20].

Previous studies [23,24,25,26] have demonstrated that transportation and lairaging under commercial conditions may result in a significant acute-phase response, since these preslaughter phases represent a complex model of stress that includes handling, loading–unloading, fasting, space restriction, and new housing. Preslaughter stress or disease activates neuroendocrinological pathways, such as the SAM and the HPA axes [52]. This activation leads to the liberation of catecholamines and glucocorticoids, for which neurotransmitters directly or indirectly (through induction of cytokines released from macrophages) initiate the acute-phase response, leading to increased synthesis of APPs in the liver. Consequently, their concentrations in circulation rapidly increase in response to positive APPs (haptoglobin, CRP, Pig-MAP, and SAA), while decrease in in response to negative APPs, such as albumin [29,52,53]. In this study, albumin level was moderately positively correlated with live weight, hot carcass weight, cold carcass weight, and back fat thickness. Additionally, loin muscle thickness and lean meat content tended to increase with increasing albumin level. When pigs are not sick or under stress (undetectable or low levels of positive APPs, and physiological level of albumin), they are able to maintain normal body homeostasis and metabolism, and thus dietary energy can be used for production processes, such as bone and muscle formation and fat deposition, leading to increases in the live and carcass weights, back fat thickness, loin muscle thickness, and carcass meatiness. However, as far as pork quality prediction is concerned, the limited correlations obtained between albumin and CRP and meat quality parameters do not indicate these physiological stress biomarkers as reliable predictors of pork quality variation at slaughter. These results were strengthened because in all pork quality classes, albumin levels were above the basal levels for slaughter pigs, while CRP levels were far below the reference values for the species. On the other hand, haptoglobin concentration was weakly positively correlated with meat pH_45min_ and pH_24h_, and weakly negatively correlated with drip loss, cooking loss, and L* value. Moreover, the highest mean haptoglobin level was recorded in DFD pork. Since synthesis of positive APPs in the liver has been associated with muscular catabolism, lower pork quality can be expected to occur in the same pig [27]. In this study, higher haptoglobin concentration weakly correlated with increased activity of CK, LDH, and AST levels, suggesting that the increased APP concentration was connected with muscular catabolism. Nevertheless, the weak correlations obtained between haptoglobin level and meat quality parameters indicate the poor reliability in predicting variation in pork quality traits. These results could be attributed to the minimal antemortem handling and standard marketing conditions resulting in stress levels that were too mild to produce the necessary variations in haptoglobin levels and pork quality characteristics. Therefore, further investigation is required to fully understand the links between APPs (haptoglobin, CRP, Pig-MAP, SAA, and albumin) and meat quality traits before any firm conclusions can be drawn about their usefulness as objective indicators of pork quality.

## 5. Conclusions

This study showed that LDH could help pork producers in prediction of pork quality variation at a time that decisions are made in the abattoir (i.e., during the period between pig slaughtering and carcass chilling). Cortisol, ALT, and albumin are of practical importance for pork producers in prediction of carcass quality variations in the slaughterline on the day of slaughter. Other examined physiological stress biomarkers had limited correlations with carcass and meat quality traits, suggesting that they are not of practical importance for pork producers in prediction of carcass and pork quality variations at slaughter. The possible reasons behind these apparent low correlations between physiological stress biomarkers and meat quality characteristics are minimal stressful preslaughter handling and standard marketing conditions, which were not sufficient to cause the high levels of stress necessary to produce variations in physiological stress biomarkers and pork quality. The repetition of this investigation under stressful or at least less-controlled conditions would presumably allow the necessary variations in physiological stress biomarkers and pork quality characteristics.

## Figures and Tables

**Table 1 animals-10-00614-t001:** Assessment of pork quality classes according to Koćwin-Podsiadła et al. [35].

Pork Quality Class	pH_24h_ ^†^	Drip Loss (%) ^‡^	L* Value ^§^
PSE meat	˂6.0	≥5	≥50
RSE meat	˂6.0	≥5	42–50
RFN meat	˂6.0	2–5	42–50
PFN meat	˂6.0	2–5	≥50
DFD meat	≥6.0	≤2	˂42

Abbreviations:^†^ pH_24h_—pH value measured 24 h postmortem; ^‡^ drip loss—fluid loss at 4 °C for a period of 24 to 72 h postmortem; ^§^ L* value—lightness. Note: pale, soft, and exudative—PSE; red, soft, and exudative—RSE; red, firm, and nonexudative—RFN; pale, firm, and nonexudative—PFN; dark, firm, and dry—DFD.

**Table 2 animals-10-00614-t002:** Correlations (r) between stress metabolites, stress hormones, electrolytes, and carcass quality traits in slaughter pigs.

	Stress Metabolites		Stress Hormones		Electrolytes	
	Lactate (mmol/L)	Glucose (mmol/L)	Cortisol (nmol/L)	ACTH (pmol/L)	Sodium (mmol/L)	Chloride (mmol/L)
LW (kg)	−0.006 (0.9522)	0.02 (0.8461)	0.36 * (0.007)	−0.18 (0.174)	0.11 (0.407)	−0.04 (0.789)
HCW (kg)	−0.006 (0.9522)	0.02 (0.8461)	0.36 * (0.007)	−0.18 (0.174)	0.11 (0.407)	−0.04 (0.789)
CCW (kg)	−0.006 (0.9522)	0.02 (0.8461)	0.36 * (0.007)	−0.18 (0.174)	0.11 (0.407)	−0.04 (0.789)
BFT (mm)	−0.16 * (0.0152)	−0.09 (0.1602)	0.49 * (<0.0001)	0.01 (0.961)	0.05 (0.604)	−0.03 (0.782)
LMT (mm)	0.16 * (0.0162)	0.09 (0.1476)	−0.18 * (0.047)	−0.16 * (0.05)	0.07 (0.438)	0.01 (0.905)
LMC (%)	0.20 * (0.0021)	0.12 * (0.05)	−0.42 * (<0.0001)	0.06 (0.542)	−0.01 (0.917)	0.05 (0.625)
SLC	0.09 (0.1649)	0.05 (0.4596)	0.26 * (0.005)	0.16 * (0.045)	0.05 (0.591)	−0.05 (0.616)

Abbreviations: LW—live weight; HCW—hot carcass weight; CCW—cold carcass weight; BFT—back fat thickness; LMT—loin muscle thickness; LMC—lean meat content; SLC—skin lesion score; ACTH—adrenocorticotropic hormone. Note: Level of significance: * *p* < 0.05; *p*-values representing differences from zero are shown in parentheses.

**Table 3 animals-10-00614-t003:** Correlations (r) between stress metabolites, stress hormones, electrolytes, and meat quality traits in slaughter pigs.

	Stress Metabolites		Stress Hormones		Electrolytes	
	Lactate (mmol/L)	Glucose (mmol/L)	Cortisol (nmol/L)	ACTH (pmol/L)	Sodium (mmol/L)	Chloride (mmol/L)
pH_45min_	−0.15 * (0.05)	−0.15 * (0.0219)	0.05 (0.557)	0.22 * (0.017)	0.07 (0.466)	0.09 (0.329)
T_45min_ (°C)	0.20 * (0.002)	0.23 * (0.0003)	−0.08 (0.396)	−0.08 (0.374)	−0.02 (0.870)	−0.10 (0.304)
pH_24h_	−0.25 * (0.0001)	−0.14 * (0.0340)	0.08 (0.408)	0.08 (0.480)	−0.003 (0.978)	0.02 (0.869)
T_24h_ (°C)	−0.07 (0.2916)	−0.09 (0.1814)	0.01 (0.233)	0.01 (0.410)	0.01 (0.988)	0.04 (0.653)
Drip loss (%)	0.30 * (<0.0001)	0.19 * (0.0041)	−0.03 (0.750)	−0.23 * (0.012)	−0.002 (0.982)	0.06 (0.530)
Thawing loss (%)	0.09 (0.1586)	0.14 * (0.0321)	−0.37 * (<0.0001)	−0.07 (0.458)	−0.003 (0.975)	0.06 (0.545)
Cooking loss (%)	0.15 * (0.0205)	0.14 * (0.0321)	−0.15 * (0.05)	−0.28 * (0.002)	0.12 (0.178)	0.17 * (0.05)
L* value	0.24 * (0.0001)	0.14 * (0.0358)	−0.10 (0.262)	−0.14 * (0.05)	−0.07 (0.484)	0.08 (0.380)
a* value	−0.18 * (0.0051)	−0.25 * (0.0001)	−0.10 (0.294)	0.20 * (0.034)	0.10 (0.293)	0.11 (0.215)
b* value	0.05 (0.4538)	0.06 (0.3468)	−0.09 (0.328)	0.17 * (0.05)	−0.003 (0.978)	−0.03 (0.759)
Sensory color	−0.22 * (0.0005)	−0.16 * (0.0124)	0.10 (0.292)	−0.11 (0.260)	0.12 (0.210)	0.14 (0.136)
Marbling	−0.05 (0.4357)	0.03 (0.6775)	0.42 ** (<0.0001)	0.17 * (0.05)	−0.09 (0.342)	−0.10 (0.279)
RM (°)	−0.18 (0.0056)	0.03 (0.6442)	−0.11 (0.330)	0.15 (0.111)	0.04 (0.708)	−0.09 (0.338)
Rigor score	0.15 * (0.0275)	0.16 * (0.0126)	−0.10 (0.269)	−0.10 (0.185)	−0.17 * (0.05)	−0.12 (0.185)

Abbreviations: pH_45min_—meat pH values measured 45 min postmortem; T_45min_—meat temperature measured 45 min postmortem; pH_24h_—meat pH values measured 24 h postmortem; T_24h_—meat temperature measured 24 h postmortem; L* value—lightness; a* value—redness; b* value—yellowness; RM—foreleg angle rigor mortis; ACTH—adrenocorticotropic hormone. Note: Level of significance: * *p* < 0.05; *p*-values representing differences from zero are shown in parentheses.

**Table 4 animals-10-00614-t004:** Differences in stress metabolites, stress hormones, and electrolytes between pork quality classes.

Pork Quality Classes	PSE Meat (n = 91)	RSE Meat (n = 30)	RFN Meat (n = 47)	PFN Meat (n = 66)	DFD Meat (n = 6)	Standard Deviation	Minimum Value	Maximum Value	Reference Values [38,39,40,41]	*p*-Value	Significance
Stress metabolites											
Lactate (mmol/L)	10.76 ^a^	8.03 ^b^	6.35 ^c^	7.57 ^b^	2.83 ^d^	5.49	2.20	24.90	0.50-5.50	<0.0001	*
Glucose (mmol/L)	10.64 ^a^	9.07 ^b^	6.35 ^c^	7.73 ^b^	3.83 ^d^	2.97	4.20	23.90	3.70-6.40	<0.0001	*
Stress hormones											
Cortisol (nmol/L)	268.00	264.10	263.30	273.40	331.40	188.70	33.61	634.20	76.00-88.00	0.8757	ns
ACTH (pmol/L)	0.75	1.48	0.54	1.02	1.50	2.61	0.22	23.04		0.7465	ns
Electrolytes											
Sodium (mmol/L)	126.5	127.30	129.10	126.9	123.80	7.44	108.00	151.00	140.00-150.00	0.5807	ns
Chloride (mmol/L)	85.08	85.35	86.76	84.27	83.00	4.76	73.00	98.00	94.00–103.00	0.3170	ns

Abbreviations: PSE meat—pale, soft, and exudative meat; RSE meat—red, soft, and exudative meat; RFN meat—red, firm, and nonexudative meat; PFN meat—pale, firm, and nonexudative meat; DFD meat—dark, firm, and dry meat; ACTH—adrenocorticotropic hormone. Note: Level of significance: * *p* < 0.05; ns: not significant (*p* > 0.05); different letters in the same row indicate a significant difference at *p* < 0.05 ^(a–d)^.

**Table 5 animals-10-00614-t005:** Correlations (r) between stress enzymes, acute-phase proteins, and carcass quality traits in slaughter pigs.

	Stress Enzymes				Acute-phase Proteins		
	CK (units/L)	LDH (units/L)	AST (units/L)	ALT (units/L)	Haptoglobin (mg/L)	CRP (mg/L)	Albumin (g/L)
LW (kg)	0.06 (0.656)	0.01 (0.968)	−0.05 (0.698)	0.49 * (<0.0001)	−0.04 (0.784)	−0.07 (0.689)	0.54 * (<0.0001)
HCW (kg)	0.06 (0.656)	0.01 (0.968)	−0.05 (0.698)	0.49 * (<0.0001)	−0.04 (0.784)	−0.07 (0.689)	0.54 * (<0.0001)
CCW (kg)	0.06 (0.656)	0.01 (0.968)	−0.05 (0.698)	0.49 * (<0.0001)	−0.04 (0.784)	−0.07 (0.689)	0.54 * (<0.0001)
BFT (mm)	0.06 (0.507)	−0.07 (0.462)	−0.06 (0.550)	0.28 * (0.002)	−0.07 (0.456)	0.02 (0.895)	0.37 * (<0.0001)
LMT (mm)	0.17 * (0.042)	0.26 * (0.004)	0.21 * (0.021)	0.32 * (<0.0001)	−0.02 (0.827)	−0.03 (0.768)	0.24 * (0.008)
LMC (%)	0.29 * (0.026)	0.16 * (0.049)	0.09 (0.313)	0.17 * (0.048)	−0.07 (0.443)	−0.09 (0.598)	0.29 * (0.001)
SLC (%)	0.27 * (0.006)	0.19 * (0.041)	0.19 * (0.041)	0.15 * (0.05)	−0.03 (0.785)	0.02 (0.814)	0.10 (0.296)

Abbreviations: LW—live weight; HCW—hot carcass weight; CCW—cold carcass weight; BFT—back fat thickness; LMT—loin muscle thickness; LMC—lean meat content; SLC—skin lesion score; CK—creatine kinase; LDH—lactic dehydrogenase; AST—aspartate amino transferase; ALT—alanine amino transferase; CRP—C-reactive protein. Note: Level of significance: * *p* < 0.05; *p*-values representing differences from zero are shown in parentheses.

**Table 6 animals-10-00614-t006:** Correlations (r) between stress enzymes, acute-phase proteins, and meat quality traits in slaughter pigs.

	Stress Enzymes				Acute-phase Proteins		
	CK (units/L)	LDH (units/L)	AST (units/L)	ALT (units/L)	Haptoglobin (mg/L)	CRP (mg/L)	Albumin (g/L)
pH_45min_	0.15 * (0.05)	0.24 * (0.009)	0.21 * (0.025)	0.07 (0.408)	0.14 * (0.05)	−0.02 (0.394)	0.10 (0.291)
T_45min_ (°C)	−0.02 (0.804)	−0.02 (0.794)	0.01 (0.925)	0.11 (0.229)	−0.05 (0.689)	−0.07 (0.451)	0.13 (0.152)
pH_24h_	0.19 * (0.05)	0.23 * (0.019)	0.06 (0.542)	0.19 * (0.038)	0.16 * (0.05)	−0.03 (0.765)	−0.04 (0.676)
T_24h_ (°C)	0.05 (0.774)	0.02 (0.898)	0.06 (0.698)	0.07 (0.699)	−0.06 (0.521)	0.03 (0.599)	0.03 (0.923)
Drip loss (%)	−0.002 (0.987)	−0.39 * (<0.0001)	0.06 (0.525)	−0.25 * (0.007)	−0.19 * (0.041)	0.03 (0.695)	−0.16 * (0.05)
Thawing loss (%)	−0.19 * (0.042)	−0.11 (0.295)	−0.19 * (0.040)	−0.22 * (0.018)	0.03 (0.772)	0.02 (0.755)	−0.39 * (<0.0001)
Cooking loss (%)	−0.14 * (0.05)	−0.40 * (<0.0001)	−0.21 * (0.024)	−0.18 * (0.05)	−0.14 * (0.0456)	0.09 (0.342)	−0.08 (0.369)
L* value	−0.22 * (0.017)	−0.14 * (0.05)	−0.05 (0.602)	−0.18 * (0.05)	−0.17 * (0.05)	−0.09 (0.338)	−0.05 (0.570)
a* value	0.12 (0.191)	0.06 (0.502)	0.08 (0.384)	0.09 (0.337)	−0.09 (0.420)	0.03 (0.736)	−0.10 (0.270)
b* value	−0.20 * (0.032)	−0.05 (0.565)	−0.04 (0.696)	0.01 (0.904)	0.10 (0.268)	−0.05 (0.566)	0.02 (0.803)
Sensory color	−0.01 (0.959)	−0.06 (0.532)	−0.10 (0.301)	0.11 (0.252)	−0.11 (0.234)	0.05 (0.586)	0.10 (0.294)
Marbling	0.03 (0.722)	−0.34 * (<0.0001)	−0.31 * (<0.0001)	0.04 (0.668)	0.10 (0.282)	−0.02 (0.752)	0.04 (0.690)
RM (°)	0.17 (0.072)	0.16 (0.081)	0.14 * (0.05)	0.18 * (0.05)	−0.20 * (0.029)	0.01 (0.908)	0.22 * (0.015)
Rigor score	0.02 (0.958)	−0.10 (0.292)	−0.16 * (0.05)	−0.04 (0.654)	0.04 (0.644)	0.01 (0.952)	−0.04 (0.662)

Abbreviations: pH_45min_—meat pH values measured 45 min postmortem; T_45min_—meat temperature measured 45 min postmortem; pH_24h_—meat pH values measured 24 h postmortem; T_24h_—meat temperature measured 24 h postmortem; L* value—lightness; a* value—redness; b* value—yellowness; RM—foreleg angle rigor mortis; CK—creatine kinase; LDH—lactic dehydrogenase; AST—aspartate amino transferase; ALT—alanine amino transferase; CRP—C-reactive protein. Note: Level of significance: * *p* < 0.05; *p*-values representing differences from zero are shown in parentheses.

**Table 7 animals-10-00614-t007:** Differences of stress enzymes and acute-phase proteins between pork quality classes.

Pork Quality Classes	PSE Meat (n = 91)	RSE Meat (n = 30)	RFN Meat (n = 47)	PFN Meat (n = 66)	DFD Meat (n = 6)	Standard Deviation	Minimum Value	Maximum Value	Reference Values [38,39,40,41]	*p*-Value	Significance
Stress enzymes											
CK (units/L)	3553.00 ^a^	3550.00 ^a^	1207.00 ^b^	3730.00 ^a^	4266.00 ^c^	1069.00	42.67	4267.00	66.00–489.00	<0.0001	*
LDH (units/L)	2691.00 ^a^	2500.00 ^a^	614.15 ^b^	2150.00 ^a^	4482.00 ^c^	1243.00	352.00	4500.00	380.00–630.00	<0.0001	*
AST (units/L)	41.45 ^a^	53.17 ^a^	40.88 ^a^	37.70 ^a^	126.00 ^b^	33.44	4.00	245.00	32.00–84.00	<0.0001	*
ALT (units/L)	32.68	34.59	37.48	36.00	39.75	11.17	10.00	61.00	31.00–58.00	0.4878	ns
Acute-phase proteins											
Haptoglobin (mg/L)	0.21 ^a^	0.19 ^a^	0.09 ^b^	0.15 ^a^	0.43 ^c^	0.29	0.08	0.90	20.00–3000.00	<0.0001	*
CRP (mg/L)	1.05	1.08	1.01	1.03	1.03	0.25	1.00	2.90	5.00–30.00	0.8953	ns
Albumin (g/L)	32.71	32.26	34.00	33.60	33.50	5.07	21.00	45.00	19.00–24.00	0.7514	ns

Abbreviations: PSE meat—pale, soft, and exudative meat; RSE meat—red, soft, and exudative meat; RFN meat—red, firm, and nonexudative meat; PFN meat—pale, firm, and nonexudative meat; DFD meat—dark, firm, and dry meat; CK—creatine kinase; LDH—lactic dehydrogenase; AST—aspartate amino transferase; ALT—alanine amino transferase; CRP—C-reactive protein. Note: Level of significance: * *p* < 0.05; ns: not significant (*p* > 0.05); different letters in the same row indicate a significant difference at *p* < 0.05 ^(a–c)^.

**Table 8 animals-10-00614-t008:** Interrelationships (r) between physiological stress biomarkers in slaughter pigs.

Stress Biomarkers	Glucose (mmol/L)	Haptoglobin (mg/L)	CRP (mg/L)	Albumin (g/L)	Sodium (mmol/L)	Chloride (mmol/L)	Cortisol (nmol/L)	ACTH (pmol/L)	CK (units/L)	LDH (units/L)	AST (units/L)	ALT (units/L)
Lactate (mmol/L)	0.21 * (0.0006)	0.06 (0.0821)	0.07 (0.4638)	−0.10 (0.2786)	−0.04 (0.6349)	−0.04 (0.6480)	−0.01 (0.9041)	0.06 (0.4995)	−0.06 (0.5120)	−0.04 (0.6779)	−0.01 (0.9379)	−0.22 * (0.0195)
Glucose (mmol/L)		−0.08 (0.4181)	−0.07 (0.6578)	−0.09 (0.3466)	0.08 (0.4145)	0.02 (0.8472)	0.05 (0.6229)	−0.07 (0.4422)	−0.07 (0.4826)	−0.12 (0.2210)	−0.07 (0.4194)	0.07 (0.4629)
Haptoglobin (mg/L)			0.04 (0.639)	0.05 (0.628)	−0.10 (0.282)	−0.03 (0.148)	0.26 * (0.012)	−0.06 (0.523)	0.14 * (0.049)	0.15 * (0.045)	0.13 * (0.05)	0.04 (0.698)
CRP (mg/L)				−0.03 (0.161)	0.10 (0.293)	0.07 (0.449)	−0.01 (0.262)	0.01 (0.272)	0.02 (0.826)	0.09 (0.599)	0.04 (0.498)	−0.02 (0.654)
Albumin (g/L)					0.50 * (<0.0001)	0.33 * (<0.0001)	0.47 * (<0.0001)	−0.02 (0.796)	0.22 * (0.018)	0.08 (0.838)	0.01 (0.959)	0.43 * (<0.0001)
Sodium (mmol/L)						0.90 * (<0.0001)	0.19 * (0.038)	−0.01 (0.960)	0.15 * (0.049)	0.08 (0.375)	0.07 (0.446)	0.19 * (0.039)
Chloride (mmol/L)							0.14 * (0.046)	0.01 (0.902)	−0.05 (0.622)	0.07 (0.423)	0.01 (0.890)	−0.01 (0.953)
Cortisol (nmol/L)								0.12 * (0.05)	0.17 * (0.047)	0.06 (0.521)	0.13 * (0.05)	0.29 * (0.001)
ACTH (pmol/L)									0.11 (0.251)	0.32 * (<0.0001)	0.28 * (0.002)	0.14 * (0.049)
CK (units/L)										0.48 * (<0.0001)	0.28 * (0.002)	0.36 * (<0.0001)
LDH (units/L)											0.60 * (<0.0001)	0.19 * (0.042)
AST (units/L)												0.27 * (0.003)

Abbreviations: CRP—C-reactive protein; ACTH—adrenocorticotropic hormone; CK—creatine kinase; LDH—lactate dehydrogenase; AST—aspartate amino transferase; ALT—alanine amino transferase. Note: Level of significance: * *p* < 0.05; *p*-values representing difference from zero are shown in parentheses.

**Table 9 animals-10-00614-t009:** Relationships between slaughter season and physiological stress biomarkers in slaughter pigs (mean value ± standard deviation).

Slaughter Season	Winter (n = 60)	Spring (n = 60)	Summer (n = 60)	Autumn (n = 60)	*p*-Value	Significance
Stress metabolites						
Lactate (mmol/L)	15.44 ± 0.56 ^a^	14.20 ± 0.69 ^b^	17.40 ±0.67 ^c^	11.58 ± 0.71 ^d^	0.048	*
Glucose (mmol/L)	8.28 ± 0.46	8.47 ± 0.27	9.39 ± 0.46 ^a^	7.72 ± 0.30 ^b^	0.0224	*
Stress hormones						
Cortisol (nmol/L)	288.30 ± 16.47 ^a^	266.70 ± 20.33 ^a^	155.80 ± 25.61 ^b^	302.20 ± 20.91 ^a^	0.0002	*
ACTH (pmol/L)	2.03 ± 0.60 ^a^	0.36 ± 0.11 ^b^	0.25 ± 0.03 ^b^	0.24 ± 0.02 ^b^	0.0046	*
Electrolytes						
Sodium (mmol/L)	122.30 ± 1.19 ^a^	127.20 ± 0.78 ^b^	137.6 ± 1.39 ^c^	124.60 ± 1.01 ^a,b^	<0.0001	*
Chloride (mmol/L)	84.48 ± 0.43 ^a^	84.04 ± 1.05 ^a^	90.04 ± 0.66 ^b^	83.35 ± 0.92 ^a^	<0.0001	*
Stress enzymes						
CK (units/L)	3634 ± 122.10 ^a^	2688 ± 305.40 ^b^	3597 ± 185.50 ^a^	2458 ± 267.80 ^b^	<0.0001	*
LDH (units/L)	2756 ±166.9 ^a^	981 ± 48.78 ^b^	2546 ± 248.20 ^a^	1105 ±87.18 ^b^	<0.0001	*
AST (units/L)	57.00 ± 3.68 ^a^	50.16 ± 11.57 ^a,b^	31.56 ± 4.03 ^b^	32.70 ± 2.33 ^b^	0.0028	*
ALT (units/L)	35.44 ± 1.23 ^a,b^	31.72 ± 2.75 ^a^	27.67 ± 1.69 ^a^	40.70 ± 2.03 ^b^	0.0004	*
Acute-phase proteins						
Haptoglobin (mg/L)	0.14 ± 0.01	0.19 ± 0.02	0.20 ± 0.03	0.20 ± 0.23	0.0859	ns
CRP (mg/L)	1.06 ± 0.06	1.00 ± 0.00	1.11 ± 0.11	1.06 ± 0.06	0.7028	ns
Albumin (g/L)	30.10 ± 0.72 ^a^	30.52 ±0.95 ^a^	37.18 ± 0.53 ^b^	32.35 ± 0.66 ^a^	<0.0001	*

Abbreviations: CRP—C-reactive protein; ACTH—adrenocorticotropic hormone; CK—creatine kinase; LDH—lactate dehydrogenase; AST—aspartate amino transferase; ALT—alanine amino transferase. Note: Level of significance: * *p* < 0.05; ns: not significant (*p* > 0.05); different letters in the same row indicate a significant difference at *p* < 0.05 ^(a–d)^.

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
