# Peer review of "Identifying Physiological Stress Biomarkers for Prediction of Pork Quality Variation"

_animals, 2020, doi:10.3390/ani10040614_

Round 1

Reviewer 1 Report

Overall, the data in this paper are interesting, provide useful data for meat quality researchers and should be published after a substantial revision. The manuscript describes a simple and easy understandable experiment with a clear objective and includes relevant literature to justify the study. The objectives are interesting and the Material and Methods section includes relevant information on the methodology used which makes it possible to replicate the experiment in future. However, the section of results and discussion must be reviewed before publication.

Simply summary

L 23: The known correlations between those physiological stress indicators and meat quality are relatively low to confirm that these parameters could be used as sensitive pork quality predictors.

Abstract

L38: Same comment as above concerning the affirmation of : be considered as sensitive predictors of pork quality variations

 Introduction

L48: Even few minutes before slaughter

L61-62: Would be interesting to add the type of impact that the cortisol concentrations had in the pork quality (e.g. higher drip loss, L colour, lower ultimate pH etc.).

L64-65:  Since the criteria for meat quality traits varies between markets, could you specify undesirable quality traits.

L65-66:  Suggest adding the information on how the imbalance of these electrolytes negatively influences pork quality (e.g. Causes pale, soft and exudative meat, etc.).

Materials and Methods

L83: Why pigs were not feed withdraw before transportation?

L84: Do the authors have some information on the handling for loading procedure?

L84: There were different compartments in the truck? If so, there was a rotation between compartments for the different transports?

L91: Please, define standard market conditions for transportation and lairage.

L94: Please, define how the PVC board was gently used to handle pigs

L95-96: How many pigs per group?

L99: Suggest removing the word apparently.

L123: Missing information on the brand, model of the scale.

L125: Missing information on the brand, model of the steel ruler.

L126-129: Please specify the exactly location where the assessment was taken (e.g. the third/fourth last rib level/ middle region of the muscle). Add this information to the text would allow comparisons between future studies.

Results

L171-283: When discussing correlation values, I suggest include in the results, whether they were strong, moderate or weak (e.g.: weak r = 0:01 to 0:39; moderate r = 0:40 to 59; and strong: r = 0.60 to 1.00.).

L171-283: The authors don’t need to give the information twice, so I suggest putting in the text just the most relevant and strong correlations, and let the other correlations only in the table.

L 232: I’m really surprised that you had six loins scored as a DFD meat, since DFD meat in pork is really hard to be seen, specially using mild stress handling conditions and short transportation. It was a very cold temperature during transportation?

Discussion

L283 – 427: I suggest indicating in the text the level of the correlations that are being discussed.

L289-293: The statement is true, but the correlations observed in this study are quite low to support this affirmation. (Lactate × pH 45min = r = 0.15;  Lactate × pH24 = r = -0.25; Glucose × pH 45min = r = 0.15;  Glucose × pH24 = r = -0.14).

L301-305: I do not agree with this affirmation, since the level of correlations found in this study is quite low to affirm that these metabolites can be used as sensitive predictor; However you can add some references from other studies to reinforce this speculation (speculation based on the results from the current study).

L322-324: The experimental design of this study does not allow this conclusion: ACTH levels and skin lesion score indicated that these physiological stress biomarkers were not only associated with carcass quality, but also with the level of stress and pre slaughter conditions.

L348-349: I suggest discussing on the correlation between Cortisol and SLC ( R = -0.26) and CK (-0.27), because the other correlation were too weak to be considered in the discussion.

L368: Did the authors assess aggressive behaviours in pigs during lairage time? If so, could be an interesting result to be added to the discussion. If not, did the authors note the lesion type on pig’s carcasses?

L368: How the authors can affirm that those lesions resulted from aggressive behaviours in the lairage and not on the farm or lesions caused by the handling?

L372-373: I agree, but this sentence needs to be revised and consistent with the results observed in this study.

L385-386: In this study, before transportation, pigs were not deprived of food and water (L83). Therefore, it cannot be used to explain the results found in this study.

Conclusion

L428-433: This section needs to be revised. The obtained results in this study, do not allow those conclusions.

Author Response

Respected Editor and Reviewer 1,

It is with excitement that we resubmit to you a revised version of manuscript “Identifying Physiological Stress Biomarkers for Prediction of Pork Quality Variations” for Animals. Thank you for giving us the opportunity to revise and resubmit this manuscript. We appreciate the time and details provided by each reviewer and by editor and have incorporated the suggested changes into the manuscript to the best of our ability. The manuscript has certainly benefited from these insightful revision suggestions. We responded specifically to each suggestion made by the reviewers in the table below. All changes are made visible by using track changes option. We look forward to working with editor and the reviewers to move this manuscript closer to publication in Animals Journal.

Sincerely,

Authors.

March 4th, 2020.

Reviewer Comments

Reviewer 1

Simply summary:

L 23: The known correlations between those physiological stress indicators and meat quality are relatively low to confirm that these parameters could be used as sensitive pork quality predictors.

Answer*

Simply summary is improved according to the obtained results (lines 26-28).

Abstract:

L38: Same comment as above concerning the affirmation of : be considered as sensitive predictors of pork quality variations

Answer*

Abstract is improved according to the obtained results (lines 55-57).

Introduction:

L48: Even few minutes before slaughter

Answer*

This sentence is removed from the manuscript (lines 67-70).

Introduction:

Would be interesting to add the type of impact that the cortisol concentrations had in the pork quality (e.g. higher drip loss, L colour, lower ultimate pH etc.).

Answer*

Added in the lines 86-92.

Introduction:

L64-65: Since the criteria for meat quality traits varies between markets, could you specify undesirable quality traits.

Answer*

Added in the lines 95-97.

Introduction:

L65-66: Suggest adding the information on how the imbalance of these electrolytes negatively influences pork quality (e.g. Causes pale, soft and exudative meat, etc.).

Answer*

Added in the lines 99-102.

Materials and Methods:

L83: Why pigs were not feed withdraw before transportation?

Answer*

In the Republic of Serbia farmers are paid according to pig body weight, which is not yet the case in many countries that continue to pay producers according to carcass quality criteria (i.e., conformation and fat scores based on the EUROP grid). That is the main reason why pigs were not feed withdraw before transportation.

Materials and Methods:

L84: Do the authors have some information on the handling for loading procedure?

Answer*

Information about handling procedure during loading is added in the lines 161-165.

Materials and Methods:

L84: There were different compartments in the truck? If so, there was a rotation between compartments for the different transports?

Answer*

Added in the lines 167-170.

Materials and Methods:

L91: Please, define standard market conditions for transportation and lairage.

Answer*

Defined in the lines 187-194.

Materials and Methods:

L94: Please, define how the PVC board was gently used to handle pigs.

Answer*

Defined in the line 198: ... gentle pressure to the pig’s hindquarters and flank...

Materials and Methods:

L95-96: How many pigs per group?

Answer*

Added in the line 196.

Materials and Methods:

L99: Suggest removing the word apparently.

Answer*

Removed from the manuscript (line 205).

Materials and Methods:

L123: Missing information on the brand, model of the scale.

Answer*

Added in the lines 232-233.

Materials and Methods:

L125: Missing information on the brand, model of the steel ruler.

Answer*

Added in the line 240.

Materials and Methods:

L126-129: Please specify the exactly location where the assessment was taken (e.g. the third/fourth last rib level/ middle region of the muscle). Add this information to the text would allow comparisons between future studies.

Answer*

Added in the lines 239-244.

Results:

L171-283: When discussing correlation values, I suggest include in the results, whether they were strong, moderate or weak (e.g.: weak r = 0:01 to 0:39; moderate r = 0:40 to 59; and strong: r = 0.60 to 1.00.).

Answer*

Added in the lines 317-324 (Materials and methods section) and in the lines 334-449 (Results section).

Results:

L171-283: The authors don’t need to give the information twice, so I suggest putting in the text just the most relevant and strong correlations, and let the other correlations only in the table.

Answer*

Corrected in the lines 334-449 (Results section).

Results:

L 232: I’m really surprised that you had six loins scored as a DFD meat, since DFD meat in pork is really hard to be seen, specially using mild stress handling conditions and short transportation. It was a very cold temperature during transportation?

Answer*

Pigs transported during winter were exposed to extremely low temperatures.

Discussion:

L283 – 427: I suggest indicating in the text the level of the correlations that are being discussed.

Answer*

Added in the lines 564-789.

Discussion:

L289-293: The statement is true, but the correlations observed in this study are quite low to support this affirmation. (Lactate × pH 45min = r = 0.15;  Lactate × pH24 = r = -0.25; Glucose × pH 45min = r = 0.15;  Glucose × pH24 = r = -0.14).

Answer*

Corrected according to the obtained results (lines 565-592).

Discussion:

L301-305: I do not agree with this affirmation, since the level of correlations found in this study is quite low to affirm that these metabolites can be used as sensitive predictor; However you can add some references from other studies to reinforce this speculation (speculation based on the results from the current study).

Answer*

Corrected according to the obtained results (lines 565-592).

Discussion:

L322-324: The experimental design of this study does not allow this conclusion: ACTH levels and skin lesion score indicated that these physiological stress biomarkers were not only associated with carcass quality, but also with the level of stress and pre slaughter conditions.

Answer*

Corrected according to the obtained results (lines 633-643).

Discussion:

L348-349: I suggest discussing on the correlation between Cortisol and SLC ( R = -0.26) and CK (-0.27), because the other correlation were too weak to be considered in the discussion.

Answer*

Corrected according to the obtained results (lines 633-643 and lines 666-675).

Discussion:

L368: Did the authors assess aggressive behaviours in pigs during lairage time? If so, could be an interesting result to be added to the discussion. If not, did the authors note the lesion type on pig’s carcasses?

L368: How the authors can affirm that those lesions resulted from aggressive behaviours in the lairage and not on the farm or lesions caused by the handling?

Answer*

We examined the lesion type on pig’s carcasses according to Faucitano (2001). Carcass skin lesions were classified as human-inflicted type bruises, fighting-type bruises and mounting-type bruises by visual assessment of shape and size to recognize their origin.

Discussion:

L372-373: I agree, but this sentence needs to be revised and consistent with the results observed in this study.

Answer*

Removed from the manuscript (lines 717-719).

Discussion:

L385-386: In this study, before transportation, pigs were not deprived of food and water (L83). Therefore, it cannot be used to explain the results found in this study.

Answer*

Corrected according to the obtained results (lines 728-737).

Conclusion:

L428-433: This section needs to be revised. The obtained results in this study, do not allow those conclusions.

Answer*

Corrected according to the obtained results (lines 796-808).

*Adzitey, F. Effect of pre-slaughter animal handling on carcass and meat quality. Int. Food Res. J. 2011, 18, 485-491.

Reference is removed from the manuscript.

Reviewer 2 Report

Line 14: Simple summary first sentence long, difficult to read. Suggest rewording or breaking sentence into two.

Line 17 to 19: “Therefore, there is a need for finding new sensitive indicators for pig welfare monitoring from the farm to the abattoir”. This sentence suggests that you are constantly monitoring the animal as it is transported from farm-gate to the abattoir. Suggest rewording. 

Line 17: “sensitive indicators”. Unsure what sensitive indicators are. Objective measurements?

Line 45: Pre-slaughter treatment. What is the treatment? Define. Do you mean transport or mixing of animals?

Line 49: Define what the standard marketing procedures are. Are they for a specific country? Who sets the marketing procedures? Does this mean not mixing unfamiliar pigs prior to transport or is this allowed? Does it mean that specific animal handling techniques?

Line 50 changes in the animal group structure. This implies animals are mixed with unfamiliar animals. Reword for greater clarity

Line 54: Detail specifics on what carcass and meat quality traits are decreased

Line 55 to 66: There is no detail as to what effect the stress metabolites have on post-mortem metabolism of meat quality. E.g. does a high or low blood lactate concentration indicate what rate of post-mortem metabolism? Is pork quality good or bad? Meat quality was influenced by cortisol – how? Good or bad, what level of cortisol? Same for CK, LDH, AST, ALT.

Line 67 Despite the fact that many physiological stress biomarkers are already in use for predicting meat quality variations. Vague. Are they used commercially? If so state exactly which ones and how they are used.

Line 69-70: Need detail on the responses of APP and how they are used as an assessment of animal welfare.

Line 72: Hypothesis and introduction weak. Evaluating stress in pigs subjected to minimal pre-slaughter stressors is not novel thus greater detailed needed as to why this experiment is necessary

Line 76. Section 2.1. Animals, pre-slaughter handling and slaughter procedure methodology. Inadequate detail in methodology.

  • How many days were the pigs housed in finishing barn? were they kept in the same groups the entire time?
  • Line 80 – use pigs not animals.
  • The methodology implies that you have 12 groups of 20 pigs (n=240) however all pre-slaughter feeding and transport was the same and they were consigned on the same day using the same driver and the same abattoir. So what are you actually testing? It appears you are simply evaluating the effect of minimal pre-slaughter stress on plasma biomarkers and meat quality which is not novel research.
  • Line 169: The individual pig was considered as an experimental unit.

Line 86 lorry waited on. Delete “on”

Line 89 but the water was. Delete “the”

Line 94: is this movement consistent with commercial standards?

Line 99 “and apparently healthy” unnecessary as have just stated are clinically normal.

Line 117 & 118: CK, AST, LDH and ALT have been previously defined so use abbreviation only.

Line 103 – 119. Poor structure to methodology, confusing to read. Improve readability. Methods must be stated sequentially. E.g. discuss that EDTA tubes were inverted immediately after it was collected. Not in the second paragraph. Why were the potassium oxalate/sodium fluoride tubes not inverted? How were the tubes stored prior to analysis on the handheld device?

Line 140-141 “The pH and temperature were measured 45 minutes and 24 hours after slaughter using a pH-meter Testo 205 (Testo AG, Lenzkirch, Germany).” Provide further detail, how was the pH and temperature measured? Via probes? Where were the probes inserted?

Line 196. Section 3.2. Very heavy and difficult to read. Recommend restructuring. Suggest breaking into more paragraphs

Line 279 Table 5 Duplication of the correlations is unnecessary

Line 287 – 289. I didn’t see any sampling of muscle glycogen in your methodology thus you cannot conclude that it is due to muscle glycogen. Line 287, “a higher lactate and glucose levels in pigs with a higher percentage of meat” does not make sense. Reword. Are you trying to say that increased eye muscle area in pigs is associated with an increased adrenaline responsiveness thus increased lactate? The major findings from the papers you have referenced to draw associations with your own work are not discussed thus is difficult to understand your argument.

Line 343. Don’t start sentence with abbreviation

Line 428 Conclusions. Weak. What is the benefit to the pig industry for your study, this is not clear? Are you suggesting that lactate, glucose, CK and LDH be measured prior to slaughter and if at risk of poor meat quality are these pigs pulled out?

Certain literature not cited;

Saco, Y., Docampo, M. J., Fabrega, E., Manteca, X., Diestre, A., Lampreave, F., & Bassols, A. (2003). Effect of transport stress on serum haptoglobin and Pig-MAP in pigs. Animal Welfare, 12(3), 403-409.

Pineiro, M., Pineiro, C., Carpintero, R., Morales, J., Campbell, F. M., Eckersall, P. D., ... & Lampreave, F. (2007). Characterisation of the pig acute phase protein response to road transport. The Veterinary Journal, 173(3), 669-674.

Author Response

Respected Editor and Reviewer 1,

It is with excitement that we resubmit to you a revised version of manuscript “Identifying Physiological Stress Biomarkers for Prediction of Pork Quality Variations” for Animals. Thank you for giving us the opportunity to revise and resubmit this manuscript. We appreciate the time and details provided by each reviewer and by editor and have incorporated the suggested changes into the manuscript to the best of our ability. The manuscript has certainly benefited from these insightful revision suggestions. We responded specifically to each suggestion made by the reviewers in the table below. All changes are made visible by using track changes option. We look forward to working with editor and the reviewers to move this manuscript closer to publication in Animals Journal.

Sincerely,

Authors.

March 4th, 2020.

Reviewer Comments

Reviewer 2

Simply summary:

First sentence long, difficult to read. Suggest rewording or breaking sentence into two.

Answer*

Sentence is removed from the manuscript (lines 14-17).

Simply summary:

Line 17 to 19: “Therefore, there is a need for finding new sensitive indicators for pig welfare monitoring from the farm to the abattoir”. This sentence suggests that you are constantly monitoring the animal as it is transported from farm-gate to the abattoir. Suggest rewording.

Line 17: “sensitive indicators”. Unsure what sensitive indicators are. Objective measurements?

Answer*

Simply summary is improved according to the obtained results (lines 19-28).

Introduction:

Line 45: Pre-slaughter treatment. What is the treatment? Define. Do you mean transport or mixing of animals?

Answer*

This sentence is changed (lines 65-67).

Introduction:

Line 49: Define what the standard marketing procedures are. Are they for a specific country? Who sets the marketing procedures? Does this mean not mixing unfamiliar pigs prior to transport or is this allowed? Does it mean that specific animal handling techniques?

Answer*

Defined in the lines 187-194.

Introduction:

Line 50 changes in the animal group structure. This implies animals are mixed with unfamiliar animals. Reword for greater clarity

Answer*

Corrected in the lines 71-72.

Introduction:

Line 54: Detail specifics on what carcass and meat quality traits are decreased

Line 55 to 66: There is no detail as to what effect the stress metabolites have on post-mortem metabolism of meat quality. E.g. does a high or low blood lactate concentration indicate what rate of post-mortem metabolism? Is pork quality good or bad? Meat quality was influenced by cortisol – how? Good or bad, what level of cortisol? Same for CK, LDH, AST, ALT.

Answer*

Added in the lines 78-111.

Introduction:

Line 67 Despite the fact that many physiological stress biomarkers are already in use for predicting meat quality variations. Vague. Are they used commercially? If so state exactly which ones and how they are used.

Answer*

Sentence is removed from the manuscript (lines 103-105).

Introduction:

Line 69-70: Need detail on the responses of APP and how they are used as an assessment of animal welfare.

Answer*

Added in the lines 105-111.

Introduction:

Line 72: Hypothesis and introduction weak. Evaluating stress in pigs subjected to minimal pre-slaughter stressors is not novel thus greater detailed needed as to why this experiment is necessary

Answer*

Improved in the lines 112-133.

Materials and Methods:

Line 76. Section 2.1. Animals, pre-slaughter handling and slaughter procedure methodology. Inadequate detail in methodology.

Answer*

Improved in the lines 136-207.

Materials and Methods:

How many days were the pigs housed in finishing barn? were they kept in the same groups the entire time?

Answer*

Added in the lines 144-146.

Materials and Methods:

Line 80 – use pigs not animals.

Answer*

Corrected in the lines 202 and 204.

Materials and Methods:

The methodology implies that you have 12 groups of 20 pigs (n=240) however all pre-slaughter feeding and transport was the same and they were consigned on the same day using the same driver and the same abattoir. So what are you actually testing? It appears you are simply evaluating the effect of minimal pre-slaughter stress on plasma biomarkers and meat quality which is not novel research.

Answer*

Added in the lines 147-158.

Materials and Methods:

Line 169: The individual pig was considered as an experimental unit.

Answer*

Corrected in the line 331.

Materials and Methods:

Line 86 lorry waited on. Delete “on”

Answer*

Corrected in the lines 175-177.

Materials and Methods:

Line 89 but the water was. Delete “the”

Answer*

Corrected in the line 183.

Materials and Methods:

Line 94: is this movement consistent with commercial standards?

Answer*

Explained in the lines 187-194.

Materials and Methods:

Line 99 “and apparently healthy” unnecessary as have just stated are clinically normal.

Answer*

Removed from the manuscript (line 205).

Materials and Methods:

Line 117 & 118: CK, AST, LDH and ALT have been previously defined so use abbreviation only.

Answer*

Full names are removed from the manuscript (lines 222-226).

Materials and Methods:

Line 103 – 119. Poor structure to methodology, confusing to read. Improve readability. Methods must be stated sequentially. E.g. discuss that EDTA tubes were inverted immediately after it was collected. Not in the second paragraph. Why were the potassium oxalate/sodium fluoride tubes not inverted? How were the tubes stored prior to analysis on the handheld device?

Answer*

Corrected in the lines 209-228.

Materials and Methods:

Line 140-141 “The pH and temperature were measured 45 minutes and 24 hours after slaughter using a pH-meter Testo 205 (Testo AG, Lenzkirch, Germany).” Provide further detail, how was the pH and temperature measured? Via probes? Where were the probes inserted?

Answer*

Corrected in the lines 255-261. Pork quality measuremnts section is improved in the lines 252-309.

Results:

Line 196. Section 3.2. Very heavy and difficult to read. Recommend restructuring. Suggest breaking into more paragraphs.

Answer*

Corrected in the lines 334-444. We divided result section on several subsections based on the examined physiological ptress biomarkers (stress metabolites, stress hormones, stress enzymes, electrolytes, acute phase proteins and interrelationships between physiological stress biomarkers).

Results:

Line 279 Table 5 Duplication of the correlations is unnecessary

Answer*

Table is corrected in the lines 444-446.

Discussion:

Line 287 – 289. I didn’t see any sampling of muscle glycogen in your methodology thus you cannot conclude that it is due to muscle glycogen. Line 287, “a higher lactate and glucose levels in pigs with a higher percentage of meat” does not make sense. Reword. Are you trying to say that increased eye muscle area in pigs is associated with an increased adrenaline responsiveness thus increased lactate? The major findings from the papers you have referenced to draw associations with your own work are not discussed thus is difficult to understand your argument.

Answer*

Corrected in the lines 565-575.

Discussion:

Line 343. Don’t start sentence with abbreviation

Answer*

Corrected in the line 662.

Conclusion:

Line 428 Conclusions. Weak. What is the benefit to the pig industry for your study, this is not clear? Are you suggesting that lactate, glucose, CK and LDH be measured prior to slaughter and if at risk of poor meat quality are these pigs pulled out?

Answer*

Corrected according to the obtained results (lines 796-808).

*Certain literature not cited;

Saco, Y., Docampo, M. J., Fabrega, E., Manteca, X., Diestre, A., Lampreave, F., & Bassols, A. (2003). Effect of transport stress on serum haptoglobin and Pig-MAP in pigs. Animal Welfare, 12(3), 403-409.

Pineiro, M., Pineiro, C., Carpintero, R., Morales, J., Campbell, F. M., Eckersall, P. D., ... & Lampreave, F. (2007). Characterisation of the pig acute phase protein response to road transport. The Veterinary Journal, 173(3), 669-674.

Answer*

Added in the lines 891-896 (References section).

**Adzitey, F. Effect of pre-slaughter animal handling on carcass and meat quality. Int. Food Res. J. 2011, 18, 485-491.

Reference is removed from the manuscript.

Reviewer 3 Report

The present paper focuses on the use of physiological stress parameters as predictive indicators of meat quality in market pigs. I think this is an important and interesting subject that should be investigated. However, overall, the organization and presentation of the data make it very difficult for the reader to interpret and summarize the major findings from the study. Therefore, the manuscript needs to be revised to clarify the most important aspects of what the authors are trying to convey.

Simple Summary:

L14-17: This is a very long sentence. It needs to be broken into multiple sentences or shortened.

L22: The different measures (LG,CK,LDH) may be too advanced for inclusion in a non-scientific simple summary.

Introduction:

My main question: Why are all of these parameters selected/included in the study? The intro reads as though they were all chosen to see if any type of stress measure would result in a significant result without tying them back to the type of stress the pigs are experiencing and the potential negative effects on meat quality. Do they all need to be included since some have already been shown to be useful for pork quality? Why not just focus on the cortisol/APPs, which have not been tested extensively?

L46: Needs to be re-written. Maybe something like "...negatively affects pig health, welfare, carcass characteristics and meat quality.

L47: This is a very long sentence. Please make into two sentences or revise.

L52: Add an "s" after "influence"

L71: What acute phase proteins did you include? Why?

Materials & Methods:

L78: "an" instead of "the" before "average live weight".

L78: Is there a standard error for the average live weight?

L83-86: More info on transport is needed. How many animals per consignment? Were there different compartments on the single deck? Were the animals kept with their previous pen mates? What brand of trailer (or other characteristics that would be helpful)? What time of year did these occur? 

L86-87: Please clarify this sentence. The Lorry waited on what for approximately 10 minutes?

L88: How large were the lairage pens? Describe these pens in more detail. What was the ground like? Where were the two nipple drinkers located? could they see through the pen or were the sides solid? What time of year did this occur?

L91: What are standard marketing conditions? Is there a document that can be cited?

L92: Remove "on the transport vehicle."

L95: Were these the same groups from the grow/finish phase or just their lairage groups? If they are not the same as the grow/finish phase, I would argue that these are not necessarily established social groups.

L99: Again, standard industry-accepted practices is a vague description and likely changes by country. Is there a document that can be cited to describe what this means?

L114-115 & L118: Include the product name in parentheses, similar to what has been done in L107 & L108. Additionally, the city should also be included with the country for each product.

L119: If these were recorded in triplicate, please report the necessary CVs.

L141: See previous comment.

Statistics

There are quite a few missing details here. Did you evaluate linearity/normality for correlations? Did you evaluate normality and variance homogeneity of the residuals? If they didn't meet assumptions, how did you deal with that in the model?

Also, I think it would help if you included a bit more on your model. Clarify that the quality classes served as your independent variable vs. biomarkers which were your dependent variables.

L161: Remove "for windows", include version info, company, and city in parentheses.

L162: Remove associations since you are evaluating correlations. 

Results:

Given the many parameters that you evaluated, I think it would be extremely beneficial to break them up into categories and structure the results descriptions and tables around those categories. For example, you have enzymes, metabolites, hormones, electrolytes, and APPs included in the study. For each group, have a separate table of correlations and a separate sub-heading within the results sections to write the description of results. This will make it much easier for the reader to interpret your findings.

The correlational descriptions are very hard to read since attention is paid to all significant results (even though many of the r values are low). The description of the results and the tables are largely redundant. They both have the same information, which should not be the case. My recommendation is to remove the r-values from the correlation results description and maybe only focus on describing significant results that have an r-value of 0.4 or above. The reader can see the other significant results (with moderate or low r values) in the tables and can make a conclusion. This may also help with focusing the discussion on the relationships that are strong instead of all significant results.

Discussion:

The discussion will need to be re-written or re-formatted once the results section is clarified and organized. Please see my comments above about re-writing the results to focus on the strong correlations in your results. This will help focus the discussion and will make it easier for the reader to understand your major conclusions. While some of the results are significant, an r-value of -0.138 is not a strong correlation and shouldn't be treated as such in the discussion. 

Additionally, there is no need to re-emphasize P-values of your results in the discussion.

Conclusion:

Please be specific here. What are pork quality variations? Are you discussing the classes, carcass characteristics, or meet quality traits? All of them together?

Table 5: Please exclude the repeated values on the opposite side of the table. This will clean it up and focus the readers attention on non-duplicated results.

Author Response

Respected Editor and Reviewer 3,

It is with excitement that we resubmit to you a revised version of manuscript “Identifying Physiological Stress Biomarkers for Prediction of Pork Quality Variations” for Animals. Thank you for giving us the opportunity to revise and resubmit this manuscript. We appreciate the time and details provided by each reviewer and by editor and have incorporated the suggested changes into the manuscript to the best of our ability. The manuscript has certainly benefited from these insightful revision suggestions. We responded specifically to each suggestion made by the reviewers in the table below. All changes are made visible by using track changes option. We look forward to working with editor and the reviewers to move this manuscript closer to publication in Animals Journal.

Sincerely,

Authors.

March 4th, 2020.

Reviewer Comments

Reviewer 3

Simply summary:

L14-17: This is a very long sentence. It needs to be broken into multiple sentences or shortened.

Answer*

Sentence is removed from the manuscript (lines 14-17).

Simply summary:

L22: The different measures (LG,CK,LDH) may be too advanced for inclusion in a non-scientific simple summary.

Answer*

I agree with You, however, I think it is impossible to replace these terms with other words.

Introduction:

My main question: Why are all of these parameters selected/included in the study? The intro reads as though they were all chosen to see if any type of stress measure would result in a significant result without tying them back to the type of stress the pigs are experiencing and the potential negative effects on meat quality. Do they all need to be included since some have already been shown to be useful for pork quality? Why not just focus on the cortisol/APPs, which have not been tested extensively?

Answer*

Explained in the lines 112-133.

Introduction:

L46: Needs to be re-written. Maybe something like "...negatively affects pig health, welfare, carcass characteristics and meat quality.

Answer*

Corrected in the lines 65-67.

Introduction:

L47: This is a very long sentence. Please make into two sentences or revise.

Answer*

This sentence is removed from the manuscript (lines 67-70).

Introduction:

L52: Add an "s" after "influence"

Answer*

Corrected in the line 73.

Introduction:

L71: What acute phase proteins did you include? Why?

Answer*

We included acute phase proteins because the data on the ways in which eleveted acute phase proteins influence the carcass and meat quality characteristics are scarce.

Materials and Methods:

L78: "an" instead of "the" before "average live weight".

Answer*

Corrected in the line 137.

Materials and Methods:

L78: Is there a standard error for the average live weight?

Answer*

Added in the line 137.

Materials and Methods:

L83-86: More info on transport is needed. How many animals per consignment? Were there different compartments on the single deck? Were the animals kept with their previous pen mates? What brand of trailer (or other characteristics that would be helpful)? What time of year did these occur?

Answer*

Explained in the lines 144-201.

Materials and Methods:

L86-87: Please clarify this sentence. The Lorry waited on what for approximately 10 minutes?

Answer*

Corrected in the lines 175-177.

Materials and Methods:

L88: How large were the lairage pens? Describe these pens in more detail. What was the ground like? Where were the two nipple drinkers located? could they see through the pen or were the sides solid? What time of year did this occur?

Answer*

Corrected in the lines 177-186.

Materials and Methods:

L91: What are standard marketing conditions? Is there a document that can be cited?

L99: Again, standard industry-accepted practices is a vague description and likely changes by country. Is there a document that can be cited to describe what this means?

Answer*

Explained in the lines 187-203.

Materials and Methods:

L92: Remove "on the transport vehicle."

Answer*

Removed (lines 195-196).

Materials and Methods:

L95: Were these the same groups from the grow/finish phase or just their lairage groups? If they are not the same as the grow/finish phase, I would argue that these are not necessarily established social groups.

Answer*

Explained in the lines 144-145 and 199-201.

Materials and Methods:

L114-115 & L118: Include the product name in parentheses, similar to what has been done in L107 & L108. Additionally, the city should also be included with the country for each product.

Answer*

Corrected in the lines 222-223 and 226-227.

Materials and Methods:

L119: If these were recorded in triplicate, please report the necessary CVs.

Answer*

Added in the line 228.

Materials and Methods:

L141: See previous comment.

Answer*

Added in the lines 301-302.

Materials and Methods:

There are quite a few missing details here. Did you evaluate linearity/normality for correlations? Did you evaluate normality and variance homogeneity of the residuals? If they didn't meet assumptions, how did you deal with that in the model?

Answer*

Added in the lines 312-314.

Materials and Methods:

Also, I think it would help if you included a bit more on your model. Clarify that the quality classes served as your independent variable vs. biomarkers which were your dependent variables.

Answer*

Added in the lines 327-328.

Materials and Methods:

L161: Remove "for windows", include version info, company, and city in parentheses.

Answer*

Removed "for windows" (line 312). Version info, company, and city are added in parentheses (lines 311-312).

Materials and Methods:

L162: Remove associations since you are evaluating correlations.

Answer*

Removed (lines 316-317).

Results:

Given the many parameters that you evaluated, I think it would be extremely beneficial to break them up into categories and structure the results descriptions and tables around those categories. For example, you have enzymes, metabolites, hormones, electrolytes, and APPs included in the study. For each group, have a separate table of correlations and a separate sub-heading within the results sections to write the description of results. This will make it much easier for the reader to interpret your findings.

The correlational descriptions are very hard to read since attention is paid to all significant results (even though many of the r values are low). The description of the results and the tables are largely redundant. They both have the same information, which should not be the case. My recommendation is to remove the r-values from the correlation results description and maybe only focus on describing significant results that have an r-value of 0.4 or above. The reader can see the other significant results (with moderate or low r values) in the tables and can make a conclusion. This may also help with focusing the discussion on the relationships that are strong instead of all significant results.

Answer*

Corrected in the lines 334-444. We divided result section on several subsections based on the examined physiological ptress biomarkers (stress metabolites, stress hormones, stress enzymes, electrolytes, acute phase proteins and interrelationships between physiological stress biomarkers).

Discussion:

The discussion will need to be re-written or re-formatted once the results section is clarified and organized. Please see my comments above about re-writing the results to focus on the strong correlations in your results. This will help focus the discussion and will make it easier for the reader to understand your major conclusions. While some of the results are significant, an r-value of -0.138 is not a strong correlation and shouldn't be treated as such in the discussion.

Answer*

Discussion section is improved according to the obtained results (lines 564-789).

Discussion:

Additionally, there is no need to re-emphasize P-values of your results in the discussion.

Answer*

Corrected throughout discussion section (lines 564-789).

Conclusion:

Please be specific here. What are pork quality variations? Are you discussing the classes, carcass characteristics, or meet quality traits? All of them together?

Answer*

Corrected according to the obtained results (lines 796-808).

*Table 5: Please exclude the repeated values on the opposite side of the table. This will clean it up and focus the readers attention on non-duplicated results.

Answer*

Table is corrected in the lines 444-446.

**Adzitey, F. Effect of pre-slaughter animal handling on carcass and meat quality. Int. Food Res. J. 2011, 18, 485-491.

Reference is removed from the manuscript.

Round 2

Reviewer 1 Report

Congratulation on the work done on reviewing this manuscript.

The information on the manuscript is more clear and well described than in the preceding version.

Regards,

Materials and Methods

L179-182: I suggest: Lairage pens had only concrete floor and walls and solid metallic gate, where  pigs  had olfactory and auditory, but not  physical or visual contact with pigs  from the other pens.  Lairage pens were not equipped with sprinkling system, thus, the ambient temperature and relative humidity at lairage pens were regulated only by natural ventilation. 

L184-185:  Don’t need to provide information on the height and location of drinkers.

L188-193:  Please, review this sentence.

Statistical analysis

L324-327:  Did you consider the season in your statistical model?

Discussion

L634-643: The processing of the carcasses on the slaughter line cold change the color of the lesions, letting it darker (brownish) but it could not affect the shape of the lesion impacting on the lesion type score. Please, review this sentence.

L681: Please, specify the exact percentage (31.25%) or the exact number of pigs.

L682-685: I suggest discussing on how the weather can impact on pigs’ behaviours and adding a reference to this sentence. Additionally, the study was performed in two different seasons; Did both seasons have the same impact on the prevalence of fighting-type bruises on pigs’ carcasses?

L683-684: Please, specify adverse weather conditions.

L733-737: Usually one nipple serves up to 15 pigs; however, there is a European recommendation for an enhanced animal welfare that proposes the ratio one nipple up to 10 pigs. Therefore, I disagree that this could lead to dehydration in pigs at this stage. I suggest to discussing on weather conditions and water deprivation before and during transportation, instead.

Author Response

Respected Editor and Reviewer 1,

It is with excitement that we resubmit to you a revised version of manuscript “Identifying Physiological Stress Biomarkers for Prediction of Pork Quality Variations” for Animals. Thank you for giving us the opportunity to revise and resubmit this manuscript. We appreciate the time and details provided by each reviewer and by editor and have incorporated the suggested changes into the manuscript to the best of our ability. The manuscript has certainly benefited from these insightful revision suggestions. We responded specifically to each suggestion made by the reviewers in the table below. All changes are made visible by using track changes option. We look forward to working with editor and the reviewers to move this manuscript closer to publication in Animals Journal.

Sincerely,

Authors.

March 27th, 2020.

Reviewer Comments

Reviewer 1

Materials and Methods:

L179-182: I suggest: Lairage pens had only concrete floor and walls and solid metallic gate, where  pigs  had olfactory and auditory, but not  physical or visual contact with pigs  from the other pens. Lairage pens were not equipped with sprinkling system, thus, the ambient temperature and relative humidity at lairage pens were regulated only by natural ventilation.

Answer*

Corrected in the lines 150-154.

Materials and Methods:

L184-185:  Don’t need to provide information on the height and location of drinkers.

Answer*

Corrected in the line 156.

Materials and Methods:

L188-193:  Please, review this sentence.

Answer*

This sentence was broken into shorter sentences (lines 160-171).

Statistical analysis

L324-327:  Did you consider the season in your statistical model?

Answer*

We included slaughter season as a random factor in statistical approach in order to control for the possible effects of ambient conditions on levels of physiological stress biomarkers (Materials and Methods section: lines 297-300 and Results section: lines 417-429).

Discussion:

L634-643: The processing of the carcasses on the slaughter line cold change the color of the lesions, letting it darker (brownish) but it could not affect the shape of the lesion impacting on the lesion type score. Please, review this sentence.

Answer*

Corrected in the lines 599-603.

Discussion:

L681: Please, specify the exact percentage (31.25%) or the exact number of pigs.

Answer*

Added in the line 675.

Discussion:

L682-685: I suggest discussing on how the weather can impact on pigs’ behaviours and adding a reference to this sentence. Additionally, the study was performed in two different seasons; Did both seasons have the same impact on the prevalence of fighting-type bruises on pigs’ carcasses?

Answer*

Added and explained in the lines 677-691.

Discussion:

L683-684: Please, specify adverse weather conditions.

Answer*

Added in the lines 677-680.

Discussion:

L733-737: Usually one nipple serves up to 15 pigs; however, there is a European recommendation for an enhanced animal welfare that proposes the ratio one nipple up to 10 pigs. Therefore, I disagree that this could lead to dehydration in pigs at this stage. I suggest to discussing on weather conditions and water deprivation before and during transportation, instead.

Answer*

Added in the lines 633-641.

Reviewer 2 Report

The revised manuscript is markedly improved.

The results section is easier to read however there are now an excessive number of tables. Your previous tables 2, 3 and 4 were good. I recommend collating tables 3,5,7,9 & 11 into one table and collating 2,4,6,8 & 10 into one table.

For tables 3,5,7,9 & 11 Differences of stress metabolites, I assume these are the mean concentrations? This needs to be stated in the table and caption. This table would benefit from also including the standard deviation, minimum and maximum concentration for each blood parameter. The table would also benefit from a column which states the published normal concentrations of each blood parameter for pigs to guide the reader as to what the difference was in the experimental pigs compared to normal concentrations.

Author Response

Respected Editor and Reviewer 2,

It is with excitement that we resubmit to you a revised version of manuscript “Identifying Physiological Stress Biomarkers for Prediction of Pork Quality Variations” for Animals. Thank you for giving us the opportunity to revise and resubmit this manuscript. We appreciate the time and details provided by each reviewer and by editor and have incorporated the suggested changes into the manuscript to the best of our ability. The manuscript has certainly benefited from these insightful revision suggestions. We responded specifically to each suggestion made by the reviewers in the table below. All changes are made visible by using track changes option. We look forward to working with editor and the reviewers to move this manuscript closer to publication in Animals Journal.

Sincerely,

Authors.

March 27th, 2020.

Reviewer Comments

Reviewer 2

The results section is easier to read however there are now an excessive number of tables. Your previous tables 2, 3 and 4 were good. I recommend collating tables 3,5,7,9 & 11 into one table and collating 2,4,6,8 & 10 into one table.

Answer*

The number of tables are significanlty reduced (from 12 to 8). However, other reviewers suggested to include slaughter season as a random factor in statistical approach (for the possible effects of ambient conditions on levels of physiological stress biomarkers), so we added one more table in the Results section (in total 9 tables).

Simply summary:

For tables 3,5,7,9 & 11 Differences of stress metabolites, I assume these are the mean concentrations? This needs to be stated in the table and caption. This table would benefit from also including the standard deviation, minimum and maximum concentration for each blood parameter. The table would also benefit from a column which states the published normal concentrations of each blood parameter for pigs to guide the reader as to what the difference was in the experimental pigs compared to normal concentrations.

Answer*

We included descriptive statistical parameters (standard deviation, and minimum and maximum range) in Table 4 (lines 349-352) and Table 7 (lines 406-409). Reference values for all physiological stress biomarkers (with the exception of the ACTH - not found in the literature) are included in Table 4 (lines 349-352) and Table 7 (lines 406-409).

Reviewer 3 Report

Simple Summary:

L20: Need to add a “to” after “order” and before “classify”

L21: Replace “to” with “for” after “carcasses”

L22-23: I suggest “…and real-time tools for predicting pork and carcass quality characteristics.”

Introduction:

Overall, this introduction is still too long. I think the information can be synthesized to shorten it. Particularly the information about the different classes of biomarkers and their specific effects on meat and carcass characteristics. The discussion is a good place to talk about the specific behaviors of certain important biomarkers as they arise in your results. In the intro, I would recommend mentioning the biomarker classes and what meat characteristics have been investigated, some general results, and the need for more research since some of the results are mixed instead of specific results for each class. The last paragraph that was added after the first revision is very good.

L66: Add “s” after “stage”

Materials & Methods:

L152-158: This is hard to read. If you are going to include this information, it would be better to summarize with a mean and SE instead of range.

L165: Include a standard error instead of range.

L167: What does “a rotation the group position” mean?

L169: This sort of clarifies my previous question but there’s no information on how many compartments are in the trailer so the readers won’t have a clear understanding of what is being described.

L172: Is it standard to not have bedding?

L176-177: Include a standard error instead of range.

L185: Remove “Then.” Commas before and after “including transportation and lairage time.”

L188-194: Good information but a very long sentence. Please break into shorter sentences.

L228: Do you have actual values for these?

L239-244: Make into two sentences.

L257: Place name of product in parentheses

L266 “…using a portable chroma meter (Product info; company, company location)…”

L286: Use brackets and parentheses to separate statement and product info.

L296: Replace e.g. with i.e.

L296: “i.e. a smaller angle…rigor mortis” should be in parentheses.

L315: “…were any correlations…”

L318-320: This is an example from the cited paper, but not a concrete finding and doesn’t need to be included here.

L320-324: This assertion is not correct. The weak correlations can’t be a tendency if they are statistically significant but weak. Please remove this.

L326: Please replace “Tukey’s post test” with the Tukey test, Tukey’s honest significant test, or Tukey procedure.

L327: Did you include any other variables or covariates in the model?

Results:

I appreciate the substantial shortening of the results and re-formatting of the tables. Much easier to read.

- Tendencies, by definition, are not significant results and shouldn’t be presented as such. If they were actually tendencies, please remove from the results and indicate in the Tables by providing an actual P-value (see my comments below in the Tables section). If these tendencies in the results section were actually significant results, but weak correlations, they should also be presented as such.

The reader should be able to make the conclusion that they are significant but weak by looking at the table, but the information needs to be communicated correctly and clearly. Presenting significant but weak correlations as tendencies is incorrect and only serves to confuse the reader.

- Please present the actual P-values in the results and/or tables so the reader can interpret the level of significance.

L433: Please present actual P-values in this section.

All Tables:

If the tendencies in the tables are true tendencies (significant above 0.05 but below 0.10), I disagree that they should be marked, since they are (by definition) not significant results and shouldn’t be treated as though they are. If you wanted to show them as tendencies (if they are true tendencies), please include the p-values in your tables instead of an asterisk. The single asterisks look like they are significant results which will confuse the reader if they are truly tendencies.

Discussion:

No need to reference specific tables in this section.

L589-590: Please reformat sentence.

L591: Replace “resulting” with “results”

L619: Correlated instead of associated

L627: Maybe “contributes to” instead of “influence”?

L639: Replace “resulting” with “results”

L643: There’s an argument to be made that the HPA-axis is not necessarily a good indicator of “chronic stress.” It takes longer to take effect compared to the SAM-axis, but cortisol typically increases within a few minutes of exposure to a stressor and, in many cases, goes back to baseline levels within a short period of time. It’s a very inconsistent measure of chronic or “long term” stress which is typically thought of in terms of days, weeks, months instead of hours.

L662: Again, these are correlations not associations. Please review this section to make sure you are using “correlated” instead of “associated” for results that were reported as correlations.

L676: Again, please be careful with language here. You can’t conclude that any variable of interest contributed to a change since only correlations were evaluated. Please make sure this mistake is not made elsewhere in the discussion.

Author Response

Respected Editor and Reviewer 3,

It is with excitement that we resubmit to you a revised version of manuscript “Identifying Physiological Stress Biomarkers for Prediction of Pork Quality Variations” for Animals. Thank you for giving us the opportunity to revise and resubmit this manuscript. We appreciate the time and details provided by each reviewer and by editor and have incorporated the suggested changes into the manuscript to the best of our ability. The manuscript has certainly benefited from these insightful revision suggestions. We responded specifically to each suggestion made by the reviewers in the table below. All changes are made visible by using track changes option. We look forward to working with editor and the reviewers to move this manuscript closer to publication in Animals Journal.

Sincerely,

Authors.

March 27th, 2020.

Reviewer Comments

Reviewer 3

Simply summary:

L20: Need to add a “to” after “order” and before “classify”

Answer*

Corrected in the lines 16.

Simply summary:

L21: Replace “to” with “for” after “carcasses”

Answer*

Corrected in the lines 16.

Simply summary:

L22-23: I suggest “…and real-time tools for predicting pork and carcass quality characteristics.”

Answer*

Corrected in the lines 17-19.

Introduction:

Overall, this introduction is still too long. I think the information can be synthesized to shorten it. Particularly the information about the different classes of biomarkers and their specific effects on meat and carcass characteristics. The discussion is a good place to talk about the specific behaviors of certain important biomarkers as they arise in your results. In the intro, I would recommend mentioning the biomarker classes and what meat characteristics have been investigated, some general results, and the need for more research since some of the results are mixed instead of specific results for each class. The last paragraph that was added after the first revision is very good.

Answer*

In the first review round, other reviewers suggested me to include some information in the introduction section about the relationship between physiological stress biomarkers and pork quality in order to improve that section.

Introduction:

L66: Add “s” after “stage”

Answer*

Corrected in the line 47.

Materials and Methods:

L152-158: This is hard to read. If you are going to include this information, it would be better to summarize with a mean and SE instead of range.

Answer*

Corrected in the lines 128-133.

Materials and Methods:

L165: Include a standard error instead of range.

Answer*

Corrected in the line 137.

Materials and Methods:

L167: What does “a rotation the group position” mean?

L169: This sort of clarifies my previous question but there’s no information on how many compartments are in the trailer so the readers won’t have a clear understanding of what is being described.

Answer*

Information about number of compartments in the transport vehicle was already included (lines 139-140).

Materials and Methods:

L172: Is it standard to not have bedding?

Answer*

Yes, it is common practise in Serbia.

Materials and Methods:

L176-177: Include a standard error instead of range.

Answer*

Corrected in the lines 146-148.

Materials and Methods:

L185: Remove “Then.” Commas before and after “including transportation and lairage time.”

Answer*

Corrected in the lines 157-158.

Materials and Methods:

L188-194: Good information but a very long sentence. Please break into shorter sentences.

Answer*

This sentence was broken into shorter sentences (lines 160-171).

Materials and Methods:

L228: Do you have actual values for these?

Answer*

Added in the lines 200-203.

Materials and Methods:

L239-244: Make into two sentences.

Answer*

Corrected in the lines 209-218.

Materials and Methods:

L257: Place name of product in parentheses

Answer*

Corrected in the line 227.

Materials and Methods:

L266 “…using a portable chroma meter (Product info; company, company location)…”

Answer*

Corrected in the lines 237-238.

Materials and Methods:

L286: Use brackets and parentheses to separate statement and product info.

Answer*

Corrected in the lines 257-259.

Materials and Methods:

L296: Replace e.g. with i.e.

Answer*

Corrected in the lines 267-268.

Materials and Methods:

L296: “i.e. a smaller angle…rigor mortis” should be in parentheses.

Answer*

Corrected in the lines 267-268.

Materials and Methods:

L315: “…were any correlations…”

Answer*

Corrected in the line 287.

Materials and Methods:

L318-320: This is an example from the cited paper, but not a concrete finding and doesn’t need to be included here.

Answer*

Deleted from the manuscript (lines 289-291).

Materials and Methods:

L320-324: This assertion is not correct. The weak correlations can’t be a tendency if they are statistically significant but weak. Please remove this.

Answer*

Deleted from the manuscript (lines 291-294).

Materials and Methods:

L326: Please replace “Tukey’s post test” with the Tukey test, Tukey’s honest significant test, or Tukey procedure.

Answer*

Corrected in the line 297.

Materials and Methods:

L327: Did you include any other variables or covariates in the model?

Answer*

We included slaughter season as a random factor in statistical approach in order to control for the possible effects of ambient conditions on levels of physiological stress biomarkers (Materials and Methods section: lines 297-300 and Results section: lines 417-429).

Results:

- Tendencies, by definition, are not significant results and shouldn’t be presented as such. If they were actually tendencies, please remove from the results and indicate in the Tables by providing an actual P-value (see my comments below in the Tables section). If these tendencies in the results section were actually significant results, but weak correlations, they should also be presented as such.

The reader should be able to make the conclusion that they are significant but weak by looking at the table, but the information needs to be communicated correctly and clearly. Presenting significant but weak correlations as tendencies is incorrect and only serves to confuse the reader.

- Please present the actual P-values in the results and/or tables so the reader can interpret the level of significance.

L433: Please present actual P-values in this section.

Answer*

Corrected in all tables and throughout results section (lines 308-429).

All Tables:

If the tendencies in the tables are true tendencies (significant above 0.05 but below 0.10), I disagree that they should be marked, since they are (by definition) not significant results and shouldn’t be treated as though they are. If you wanted to show them as tendencies (if they are true tendencies), please include the p-values in your tables instead of an asterisk. The single asterisks look like they are significant results which will confuse the reader if they are truly tendencies.

Answer*

We added exact P values in all tables throughout results section (lines 308-429).

Since one of the reviewers suggesed to include reference values and descriptive statistical parameters (standard deviation, and minimum and maximum range) in Table 4 (lines 349-352) and Table 7 (lines 406-409) and that revised manuscript has an excessive number of tables, we rearangged tables and reduced their number (from 12 to 9). We hope that result section and tables are still clear and easy to read.

Discussion:

No need to reference specific tables in this section.

Answer*

Corrected throughout discussion section (lines 546-734).

Discussion:

L589-590: Please reformat sentence.

Answer*

Corrected in the lines 573-575.

Discussion:

L591: Replace “resulting” with “results”

L639: Replace “resulting” with “results”

Answer*

Corected in the lines 576 and 596.

Discussion:

L619: Correlated instead of associated

Answer*

Corrected throughout discussion section (lines 546-734).

Discussion:

L643: There’s an argument to be made that the HPA-axis is not necessarily a good indicator of “chronic stress.” It takes longer to take effect compared to the SAM-axis, but cortisol typically increases within a few minutes of exposure to a stressor and, in many cases, goes back to baseline levels within a short period of time. It’s a very inconsistent measure of chronic or “long term” stress which is typically thought of in terms of days, weeks, months instead of hours.

Answer*

Added in the lines 626-630.

Discussion:

L662: Again, these are correlations not associations. Please review this section to make sure you are using “correlated” instead of “associated” for results that were reported as correlations.

L676: Again, please be careful with language here. You can’t conclude that any variable of interest contributed to a change since only correlations were evaluated. Please make sure this mistake is not made elsewhere in the discussion.

Answer*

Corrected throughout discussion section (lines 546-734).